# Graph Potential Field Neural Network for Massive Agents Group-wise Path Planning

**Yueming Lyu**[*]                                                                  *Lyu_Yueming@cfar.a-star.edu.sg*
*Centre for Frontier AI Research*
*Institute of High Performance Computing*
*Agency for Science, Technology and Research, Singapore*

**Xiaowei Zhou**                                                                    *zhouxiaowei@ouc.edu.cn*
*College of Computer Science and Technology*
*Ocean University of China*

**Xingrui Yu**                                                                      *yu_xingrui@cfar.a-star.edu.sg*
*Centre for Frontier AI Research*
*Institute of High Performance Computing*
*Agency for Science, Technology and Research, Singapore*

**Ivor W. Tsang**                                                                   *Ivor_Tsang@cfar.a-star.edu.sg*
*Centre for Frontier AI Research*
*Institute of High Performance Computing*
*Agency for Science, Technology and Research, Singapore*
*College of Computing and Data Science, Nanyang Technological University, Singapore*

**Reviewed on OpenReview:** *https://openreview.net/forum?id=LJHVPWNnV6*

## Abstract

Multi-agent path planning is important in both multi-agent path finding and multi-agent reinforcement learning areas. However, continual group-wise multi-agent path planning that requires the agents to perform as a team to pursue high team scores instead of individually is less studied. To address this problem, we propose a novel graph potential field-based neural network (GPFNN), which models a valid potential field map for path planning. Our GPFNN unfolds the T-step iterative optimization of the potential field maps as a T-layer feedforward neural network. Thus, a deeper GPFNN leads to more precise potential field maps without the over-smoothing issue. A potential field map inherently provides a monotonic potential flow from any source node to the target nodes to construct the optimal path (w.r.t. the potential decay), equipping our GPFNN with an elegant planning ability. Moreover, we incorporate dynamically updated boundary conditions into our GPFNN to address group-wise multi-agent path planning that supports both static targets and dynamic moving targets. Empirically, experiments on three different-sized mazes (up to $1025 \times 1025$ sized mazes) with up to 1,000 agents demonstrate the planning ability of our GPFNN to handle both static and dynamic moving targets. Experiments on extensive graph node classification tasks on six graph datasets (up to millions of nodes) demonstrate the learning ability of our GPFNN.

## 1 Introduction

Multi-agent navigation and tracking on a graph is a challenging problem in the field of artificial intelligence, machine learning and robotics, as it requires coordination and cooperation among multiple agents to achieve a common goal while avoiding collisions. In this problem, multiple agents corporately move on the graph and track their (static or dynamic) targets while avoiding collisions with each other (Sharon et al., 2015; Stern et al., 2019; Lin et al., 2022; Surynek, 2022). The goal is to reduce the total moving cost while catching

---

[*]Corresponding author

as many targets as possible. It has many practical applications, such as in warehouse management, traffic control, unmanned aerial vehicles (UAVs) navigation, and package delivery (Lin et al., 2022; Surynek, 2022).

The multi-agent pathfinding (MAPF) is a key subarea of multi-agent navigation and tracking on graphs. It focuses on constructing a valid path for each agent to its targets without collisions (Surynek, 2022; Felner et al., 2017). One of the main limitations of current research on the multi-agent tracking problem is scalability. The multi-agent pathfinding problem is known to be NP-hard, and many of the existing algorithms for solving it have exponential time complexity in the worst case (Sharon et al., 2015; Goldenberg et al., 2014; Gordon et al., 2021). This means that these algorithms may not be able to handle large-scale problems with many agents or complex environments.

Moreover, most of the current research on multi-agent pathfinding focuses on solving path planning for pre-assigned one-to-one agent-target pairs (Sharon et al., 2015; Goldenberg et al., 2014; Gordon et al., 2021; Tang et al., 2024). However, in practice, many situations require the agents to perform as a team instead of individually. Moreover, it requires the agents continually to search the remaining targets instead of stopping after reaching a target, e.g., package delivery from the same company. As a result, there is an increasing demand for continual group-wise multi-agent path planning. Current methods designed for one-to-one MAPF cannot be easily extended to continual group-wise multi-agent path planning because of their exponential time complexity and the requirement of an additional target assignment procedure. Finding the optimal target assignment itself is a very challenging combination optimization problem. In addition, in real-world applications, the targets can move over time, and the agents may need to adapt their paths accordingly. Many of the existing algorithms (Sharon et al., 2015; Goldenberg et al., 2014; Tang et al., 2024; Felner et al., 2017; Surynek, 2022; Lin et al., 2022) for MAPF assume a static target, which can limit their applicability in practical settings.

To address the continual group-wise multi-agent path planning problem, we propose a novel graph potential field neural network method. Our method builds upon the graph potential field (Masoud, 2008). In contrast to the scalar value formulation for static planning from the start node to the target node in (Masoud, 2008), we first formulate the graph potential field in a matrix form to handle multi-agents explicitly. Based on our matrix formulation, we develop our graph potential field neural network that unfolds the iterative optimization of the potential field as a T-layer feedforward neural network. Furthermore, we employ a nonlinear log-sum-exp aggregation as the nonlinear activation function instead of a linear aggregation. This not only reduces the numeric error but also increases the learning ability of our GPFNN. Moreover, we incorporate dynamically updated boundary conditions into our GPFNN to address group-wise multi-agent path planning that supports both static targets and dynamic moving targets. In addition, we further investigate the learning ability of our GPFNN on node classification tasks.

Our contributions are summarized as follows:

- To the best of our knowledge, we developed the first graph potential field-based neural network (GPFNN) with planning ability. Our GPFNN unfolds a T-step iterative optimization of the potential field maps as a T-layer feedforward neural network. Thus, a deeper GPFNN leads to more precise potential field maps. A valid potential field map provides a monotonic potential flow from any source node to the target nodes to construct the optimal path (w.r.t. the potential decay), equipping our GPFNN with a planning ability. Moreover, we incorporate dynamically updated boundary conditions into our GPFNN to address group-wise multi-agent path planning that supports both static targets and dynamic moving targets.

- Our GPFNN provides a general framework for both graph path planning tasks and graph learning tasks (e.g., node classification), which demonstrates both the planning ability and learning ability. This may suggest that our GPFNN may have the potential to serve as a useful neural network architecture for downstream tasks that require both planning ability and learning ability, e.g., multi-agent reinforcement learning.

- Empirically, extensive experiments with up to $1,000$ agents on up to $1025 \times 1025$ large maps with both static targets and dynamic targets demonstrate the planning ability of our GPFNN to handle

group-wise multi-agent path planning. Moreover, extensive experiments on node classification tasks up to millions of nodes demonstrate the learning ability of our GPFNN.

## 2 Related Works

### 2.1 Multi-agent Navigation and Tracking on Graph

The goal of multi-agent navigation and tracking on a graph is to find a set of collision-free paths for multiple agents to reach their respective goal locations while avoiding collisions with each other and any obstacles in the environment. Multi-agent navigation and tracking on a graph has applications in various domains, including robotics, transportation, and logistics. The problem is challenging due to the combinatorial nature of the search space and the need to ensure that the solutions satisfy complex constraints and objectives.

#### 2.1.1 Multi-agent Pathfinding (MAPF)

The multi-agent pathfinding (MAPF) is a key subarea of multi-agent navigation and tracking. The multi-agent pathfinding problem involves finding a collision-free path for a group of agents from their initial positions to their respective goal positions while avoiding collisions with each other. This problem is known to be NP-hard and is of practical importance in domains such as robotics, transportation, and logistics.

Recent advances in this field have led to the development of efficient algorithms for solving multi-agent navigation and tracking on a graph problems, such as Conflict-Based Search (CBS) (Sharon et al., 2015; Boyarski et al., 2015; Barer et al., 2014), Enhanced Partial Expansion A* (EPEA*) (Goldenberg et al., 2014), and Independence Detection (ID) (Silver, 2005).

Conflict-Based Search (CBS) (Sharon et al., 2015) is a widely used algorithm for solving the multi-agent pathfinding problem (MAPF). However, it has some limitations that can affect its performance in certain scenarios. CBS has a worst-case time complexity that is exponential in the number of agents, which can make it computationally infeasible for large-scale problems with many agents. This severely limits the practical usage of CBS for large-scale multi-agent navigation and tracking on a graph. Moreover, CBS can not handle the MAPF with dynamic targets.

Surynek (2021) introduce a multi-goal multi agent path finding method by extending CBS. Specifically, the approach is based on the decoupling of the problem into two stages: a goal vertex ordering stage, where the ordering of goal vertices is determined, and a pathfinding stage, where paths for each agent are computed based on the determined ordering. Despite its success to some extent, it is relying on CBS, which is not salable to large-scale multi-goal multi agent path finding problems.

Some recent works try to address different setting of MAPF. Ma et al. (2018) propose to address time-constrained MAPF problems. Dergachev & Yakovlev (2022) try to address MAPF problems with different sized agents. Švancara et al. (2019) and Ma (2021) study the online MAPF problems. For a more detailed recent survey of MAPF, one can check (Surynek, 2022; Lin et al., 2022). In addition, one may check (Stern et al., 2019) for different setting and variants of MAPF problems.

A closely related subarea is the target assignment and path finding (TAPF) (Ma & Koenig, 2016). Different from the pre-determined agent-target assignment in MAPF, TAPF considers the anonymous agents case (also known as the unlabeled MAPF) that performs the assignment of each target to each agent to minimize the cost (Ma & Koenig, 2016; Hoenig et al., 2018; Yu & LaValle, 2012). Along this line, Ma & Koenig (2016) study the TAPF problem such that agents are partitioned into teams. Each team is given the same number of unique targets (goal locations) as there are agents in the team. Each agent has to move to exactly one target given to its team such that all targets are visited. A conflict-based min-cost-flow algorithm is proposed to minimize the cost. Hoenig et al. (2018) propose to solve TAPF by leveraging K-best target assignments to create multiple search trees and employing Conflict-Based Search to resolve collisions in each tree, the proposed method is named as CBS-TA. The authors further extend CBS-TA and provide a suboptimal method, called ECBS-TA, for fast approximation. Recently, Tang et al. (2023) argue that CBS-TA suffers from poor scalability as the number of agents or targets increases. An incremental Target Assignment CBS

algorithm is proposed for acceleration. Besides the above methods, Okumura & Défago (2021) studied the offline and online TAPF, and proposed a sub-optimal algorithm, named as TSWAP, for fast computation.

Although these methods achieve good performance for TAPF problems, the problems they studied are different from our focus. The TAPF setting requires the agent to stop once it reaches the target. In contrast, we focus on the continual group-wise planning problems in which the agents that achieve (reach) the target can continue to search for the remaining targets. Moreover, our method supports handling dynamic moving targets, while most TAPF methods cannot.

Besides the research on MAPF and TAPF, many potential field based methods have been proposed for robot navigation in physical 2D-Euclidean space (Barraquand et al., 1992; Wray et al., 2016; Daily & Bevly, 2008b; Wang & Chirikjian, 2000; Masoud, 2012). In particular, the harmonic potential field methods take advantage of the property of no local optima for path planning (Kim & Khosla, 1991; Connolly et al., 1990; Daily & Bevly, 2008a; Panati et al., 2015; Szulczyński et al., 2011) However, few of them are developed for graph planning (Masoud, 2008). More importantly, how to explore multi-agent graph planning is still unexplored.

## 2.2 Graph Neural Networks

Recently, graph neural networks (GNN) (Wu et al., 2020) have emerged as state-of-the-art models for graph representation learning, e.g. convolutional graph neural networks (Kipf & Welling, 2017; Veličković et al., 2018) for node classification tasks. Spectral-based convolutional graph neural network is one important types of convolutional graph neural networks. Spectral-based convolutional neural networks develop from graph signal processing (Shuman et al., 2013). The key idea of this stream of methods is performing convolution in spectral domain. One typical spectral-based method is graph convolutional network (GCN)(Kipf & Welling, 2017). It uses the first-order approximation to define graph convolution operation, which is fast and scalable on graphs.

However, over-smoothing (Li et al., 2018; Oono & Suzuki, 2019; Chen et al., 2020; Huang et al., 2020) is found as a common issue with graph neural networks (GNNs). Namely, as the depth of GNN grows, the feature of each node tends to be the same. Some methods have been proposed to tackle this issue from different perspectives. Upon vanilla GCN, GCNII (Chen et al., 2020) solves over-smoothing by adding a skip connection from the input at each layer and adding an identity matrix to the weight matrix. RevGNN (Li et al., 2021) employs reversible connections in the design of deep GNNs, which are free of memory consumption bottlenecks. (Xu et al., 2018) proposes the jumping knowledge network to adaptively concatenate the learned representations from each intermediate layer. (Rong et al., 2020) proposes IncepGCN that borrows the inception (Szegedy et al., 2016) network structure in computer vision into GNNs model design. Recently, Li et al. (2019; 2020b); Dai et al. (2022); Zhou et al. (2024) employ the GNN to address the MAPF problems. Unlike these methods, our work is focused on group-wise MAPF problems. Additionally, our GPFNN is capable of handling dynamic targets, whereas existing GNN-based methods are limited to static targets. Furthermore, our GPFNN can scale to highly complex 1025x1025 mazes with dense obstacle configurations, a level of scalability that the experiments in these prior works have not achieved.

# 3 Graph Potential Field Neural Network

## 3.1 Graph Potential Field

In this section, we first introduce the graph potential field in (Masoud, 2008). Then, we formulate the graph potential field in a matrix form to handle multi-agents. Then, we show how to construct valid potential field features of all nodes on the graph by adding virtual boundary nodes.

Given an undirected graph $\mathcal{G} = (\mathcal{V}, \mathcal{E})$. Let $c_{ij}$ denote the cost weight associated with edge $e_{ij} \in \mathcal{E}$ (the cost of moving between node $i$ and node $j$ ). Suppose $c_{ij} > 0$ for $\forall e_{ij} \in \mathcal{E}$. Let $u_i$ denote the potential associated with node $i$ for $1 \leq i \leq |\mathcal{V}| = n$. The balance condition is defined as

$$\sum_{j \in \mathcal{N}(i)} \frac{u_i - u_j}{c_{ij}} = 0, \tag{1}$$

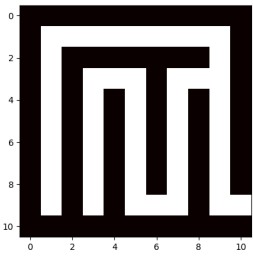

(a) 11x11-sized Maze

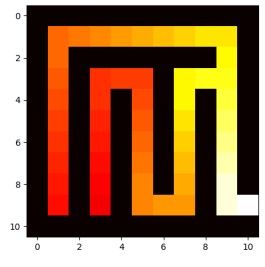

(b) Potential Field Map on the Maze. The brighter the color is, the closer to the exit.

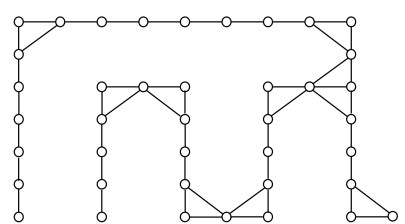

(c) An Equivalent Graph

Figure 1: Illustration of a $11 \times 11$-sized maze, the potential field constructed by our GPFNN, and an equivalent graph of the maze.

where $\mathcal{N}(i)$ denote the neighbour of node $i$, i.e., $\mathcal{N}(i) := \{j | \exists e_{ij} \in \mathcal{E}\}$.

We first formulate the potential field in a matrix form. Let the matrix $\boldsymbol{C}$ with element $\boldsymbol{C}_{ij} = \frac{1}{c_{ij}}$, and the diagonal matrix $\overline{\boldsymbol{D}}$ with diagonal element $\overline{\boldsymbol{D}}_{ii} = \sum_{j \in \mathcal{N}(i)} \frac{1}{c_{ij}}$. A generalized graph Laplacian matrix can be defined as

$$\overline{\boldsymbol{L}} = \boldsymbol{I} - \overline{\boldsymbol{D}}^{-1}\boldsymbol{C}. \tag{2}$$

The balance condition of graph potential filed in a matrix form is given in Eq.(3)

$$\overline{\boldsymbol{L}}\boldsymbol{u} = \left(\boldsymbol{I} - \overline{\boldsymbol{D}}^{-1}\boldsymbol{C}\right)\boldsymbol{u} = 0. \tag{3}$$

We can see that Eq.(3) has a trivial solution $\boldsymbol{u} = u\boldsymbol{1}$, which is the key reason of the over-smoothing issue of GNNs to construct a non-degenerate potential field. (We will discuss this with more details in section 3.3)

To get a non-degenerate graph potential field for path planning, it needs to satisfy the boundary condition besides the balance condition. We associate a virtual node $\widehat{v}_i$ with each node $i$ in the original graph $\mathcal{G}$. Let $\mathcal{O}, \mathcal{T}$ denote the set of virtual obstacle nodes and virtual target nodes, respectively. The boundary condition for all virtual nodes is

$$\widehat{u}_i = \begin{cases} b & \text{if the virtual node } \widehat{v}_i \text{ belongs to the obstacle set } \mathcal{O} \\ 0 & \text{if the virtual node } \widehat{v}_i \text{ belongs to the target set } \mathcal{T} \end{cases} \tag{4}$$

where $b > 0$ is an upper bound of the potential, i.e., $\forall u_i \in [0, b]$ for $i \in \{1, \cdots, n\}$ .

Combine both boundary condition and balance condition, we get

$$u_i = \begin{cases} \sum_{j \in \mathcal{N}(i)} w_{ij} u_j + \lambda_i b & \text{if } \widehat{v}_i \in \mathcal{O} \\ \sum_{j \in \mathcal{N}(i)} w_{ij} u_j & \text{if } \widehat{v}_i \in \mathcal{T} \end{cases}, \tag{5}$$

where $w_{ij} = \frac{\frac{1}{c_{ij}}}{1 + \sum_{j \in \mathcal{N}(i)} \frac{1}{c_{ij}}}$ is the normalized diffusion weight, and $\lambda_i = \frac{1}{1 + \sum_{j \in \mathcal{N}(i)} \frac{1}{c_{ij}}}$ is the weight associated with the virtual node. We can see that $\lambda_i + \sum_{j \in \mathcal{N}(i)} w_{ij} = 1$.

We formulate the above conditions of a non-degenerate graph potential field in a matrix form as

$$(\boldsymbol{I} - \widehat{\boldsymbol{D}}^{-1}\boldsymbol{C})\boldsymbol{u} = \boldsymbol{\lambda} \odot \widehat{\boldsymbol{u}}, \tag{6}$$

where $\widehat{\boldsymbol{D}}$ is a diagonal matrix with diagonal element $\widehat{\boldsymbol{D}}_{ii} = 1 + \sum_{j \in \mathcal{N}(i)} \frac{1}{c_{ij}}$, vector $\boldsymbol{\lambda} = [\lambda_1, \cdots, \lambda_n]^\top$ denotes the diffusion weight of virtual nodes, and vector $\widehat{\boldsymbol{u}} = [\widehat{u}_1, \cdots, \widehat{u}_n]^\top$ denotes the potential of virtual boundary nodes , and $\odot$ denotes the element-wise product operation.

According to the property of graph potential field in (Masoud, 2008), given a non-degenerate graph potential field, we can find a global minimum of potential (target node) from any source node greedily without suffering from local optimum.

## 3.2 Architecture of GPFNN

In this section, we first present a general architecture form of our GPFNN. Then, we show a particular form of our GPFNN has a linear rate of convergence to the valid potential field maps.

Our GPFNN have a general architecture form as Eq.(7)

$$\boldsymbol{U}^{t+1} = \sigma\left((1-\alpha)\boldsymbol{U}^t\boldsymbol{W}_1^t + \alpha\left(\boldsymbol{M}\boldsymbol{U}^t + \boldsymbol{\Lambda}\odot\widehat{\boldsymbol{U}}\right)\boldsymbol{W}_2^t\right),\tag{7}$$

where $\widehat{\boldsymbol{U}} = [\widehat{\boldsymbol{u}}_1, \cdots, \widehat{\boldsymbol{u}}_d] \in \mathbb{R}^{n\times d}$ denotes the potential of the virtual boundary nodes in $d$ virtual maps , and $\boldsymbol{\Lambda} = [\boldsymbol{\lambda}_1, \cdots, \boldsymbol{\lambda}_d]$ denotes the diffusion weights associated with the virtual nodes. and $\boldsymbol{M} = \widehat{\boldsymbol{D}}^{-1}\boldsymbol{C}$ denotes the propagation matrix. And $\boldsymbol{W}_1^t, \boldsymbol{W}_2^t \in \mathbb{R}^{d\times m}$ denotes trainable weight matrix at $t^{th}$ layer (it can be shared across layers). The $\sigma(\cdot)$ denote an activation function, and $\alpha$ is a hyper-parameter such that $0 \le \alpha \le 1$.

It is worth noting that each column vector $\widehat{\boldsymbol{u}}_j$ in $\widehat{\boldsymbol{U}}$ can be constructed by randomly assigning the type of virtual nodes ($\widehat{v} \in \mathcal{O}$ or $\widehat{v} \in \mathcal{T}$) or by design based on domain knowledge. Moreover, the sparse diffusion weight $w_{ij}$ w.r.t. each edge in $\boldsymbol{M}$ can either be learned or fixed as a prior.

Now, we show a particular form of GPFNN for path planning, To construct a valid potential field map, we set $\boldsymbol{W}_1^t = \boldsymbol{W}_2^t = \boldsymbol{I}$ for $t \in \{1, \cdots, T\}$, and set $\sigma(\cdot)$ as Identity map, then we obtain a simple feed-forward NN architecture as Eq.(8):

$$\boldsymbol{U}^{t+1} = (1-\alpha)\boldsymbol{U}^t + \alpha\left(\boldsymbol{M}\boldsymbol{U}^t + \boldsymbol{\Lambda}\odot\widehat{\boldsymbol{U}}\right).\tag{8}$$

We can see that Eq.(8) is the Richardson update with a step-size $\alpha$ for solving linear equation (9).

$$\left(\boldsymbol{I} - \widehat{\boldsymbol{D}}^{-1}\boldsymbol{C}\right)\boldsymbol{U} = \boldsymbol{\Lambda}\odot\widehat{\boldsymbol{U}}.\tag{9}$$

Let $\boldsymbol{U}^*$ denote the solution of the linear equation 9. According to the property of Richardson update (Hackbusch, 1994), we know that for $T$-step update (a $T$-layered NN), we can achieve the following linear rate of convergence:

$$\begin{aligned}\|\boldsymbol{U}^{T+1} - \boldsymbol{U}^*\|_F &\le (\|(1-\alpha)\boldsymbol{I} + \alpha\widehat{\boldsymbol{D}}^{-1}\boldsymbol{C}\|_2)^T\|\boldsymbol{U}^1 - \boldsymbol{U}^*\|_F\\ &\le e^{-(1-\rho)T}\|\boldsymbol{U}^1 - \boldsymbol{U}^*\|_F,\end{aligned}\tag{10}$$

where $\|\cdot\|_2$ and $\|\cdot\|_F$ denote the spectral norm and Frobenius norm, respectively. And $\rho = \|(1-\alpha)\boldsymbol{I} + \alpha\widehat{\boldsymbol{D}}^{-1}\boldsymbol{C}\|_2$. When $\rho = \|(1-\alpha)\boldsymbol{I} + \alpha\widehat{\boldsymbol{D}}^{-1}\boldsymbol{C}\|_2 < 1$, Eq.(10) shows a linear rate of convergence.

It is worth remarking that despite the simple NN architecture in Eq.(8), we can still learn the diffusion parameter $w_{ij}$ in $\boldsymbol{M}$ to adjust the cost weights associated with the edges in graph $\mathcal{G}$. This can adjust the distance between nodes because even nodes connected by an edge may belong to different classes. A nice property of this simple neural network is that it maintains the linear convergence to $\boldsymbol{U}^*$ despite the learning parameters in $\boldsymbol{M}$.

## 3.3 Relationship with current GNNs

In this section, we compare our GPFNN with current GNNs, and show that a related form of current GNNs converge linearly to a trivial feature map. In contrast, our GPFNN converge linearly to a valid potential field map by satisfying boundary conditions.

Given an undirected graph $\mathcal{G} = (\mathcal{V}, \mathcal{E})$ with adjacency matrix $\boldsymbol{A}$ and diagonal degree matrix $\boldsymbol{D}$. The GNN layer has a general form (may have other similar variants (Wu et al., 2020)) as follows:

$$\boldsymbol{U}^{t+1} = \sigma\left((1-\alpha)\boldsymbol{U}^t\boldsymbol{W}_1^t + \alpha\boldsymbol{D}^{-1}\boldsymbol{A}\boldsymbol{U}^t\boldsymbol{W}_2^t\right).\tag{11}$$

When we set $\boldsymbol{W}_1^t = \boldsymbol{W}_2^t = \boldsymbol{I}$ for $t \in \{1, \cdots, T\}$, and set $\sigma(\cdot)$ as Identity map same as the setting of our GPFNN, then we have the following NN architecture:

$$\boldsymbol{U}^{t+1} = (1-\alpha)\boldsymbol{U}^t + \alpha\boldsymbol{D}^{-1}\boldsymbol{A}\boldsymbol{U}^t, \tag{12}$$

which is the Richardson update for solving linear equation (13).

$$\left(\boldsymbol{I} - \boldsymbol{D}^{-1}\boldsymbol{A}\right)\boldsymbol{U} = \boldsymbol{0}. \tag{13}$$

We can see that $\overline{\boldsymbol{U}} = \boldsymbol{1}\boldsymbol{z}^\top$ is a solution of Eq.(13), which means that all embedding feature vectors are same, i.e., $\boldsymbol{z}_1 = \cdots = \boldsymbol{z}_n = \boldsymbol{z}$. According to the property of Richardson update (Hackbusch, 1994), we know it converge linearly to the trivial solution $\overline{\boldsymbol{U}} = \boldsymbol{1}\boldsymbol{z}^\top$ as follows

$$\|\boldsymbol{U}^{T+1} - \overline{\boldsymbol{U}}\|_F \le (\|(1-\alpha)\boldsymbol{I} + \alpha\boldsymbol{D}^{-1}\boldsymbol{A}\|_2)^T \|\boldsymbol{U}^1 - \overline{\boldsymbol{U}}\|_F \tag{14}$$
$$\le e^{-(1-\overline{\rho})T}\|\boldsymbol{U}^1 - \overline{\boldsymbol{U}}\|_F,$$

where $\overline{\rho} = \|(1-\alpha)\boldsymbol{I} + \alpha\boldsymbol{D}^{-1}\boldsymbol{A}\|_2$. This shows a over-smoothing issue in current GNNs for constructing a valid potential field map.

Compared our GPFNN in Eq.(8) with GNNs in Eq.(11), our GPFNN includes a boundary condition on virtual nodes, which is the key to solve the over-smoothing issue. Moreover, our GPFNN can learn the diffusion weights to handle weighted graph, while maintains the linear rate of convergence to a valid potential field map.

### 3.4 Optimization Property for GPFNN with Nonlinear Aggregation

In previous subsection, we discuss the optimization property for GPFNN with linear activation $\sigma(\cdot)$. In this section, we show how our GPFNN maintain the optimization property with a nonlinear aggregation.

To be simple, we assume the the cost $c_{ij} = 1$ for all edges $e_{ij} \in \mathcal{E}$. Then, we have $\lambda_i = \frac{1}{1+d_i}$, where $d_i$ denotes the degree of the node $i$.

When fix the weight matrix $\boldsymbol{W}_1^t = \boldsymbol{W}_2^t = \boldsymbol{I}$, the update rule for node $i$ with linear activation is given as follows

$$u_i^{t+1} = (1-\alpha_i)u_i^t + \frac{\alpha_i}{1+d_i}\sum_{j \in \mathcal{N}(i)} u_j^t + \frac{\alpha_i}{1+d_i}\widehat{u}_i. \tag{15}$$

By setting an adaptive step size $\alpha_i$ for each node $i$ as

$$\alpha_i = \frac{1+d_i}{2+d_i}, \tag{16}$$

we can achieve that

$$u_i^{t+1} = \frac{u_i^t + \sum_{j \in \mathcal{N}(i)} u_j^t + \widehat{u}_i}{2+d_i}. \tag{17}$$

Let $q_i = \log(u_i)$ be a mapping between $q_i$ and $u_i$ for all nodes $i \in \{1, \cdots, n\}$, then we have

$$q_i^{t+1} = \log\left(\frac{\exp(q_i^t) + \sum_{j \in \mathcal{N}(i)} \exp(q_j^t) + \exp(\widehat{q}_i)}{2+d_i}\right). \tag{18}$$

The connection between update rule in Eq.(17) and update rule in Eq.(18) is shown in Proposition 1. The proof of Proposition 1 is presented in the Appendix A.

**Proposition 1.** *Let the boundary condition of potential in original space be* $\widehat{u}_i = \begin{cases} 1 & \text{if } i \in \mathcal{O} \\ 0 & \text{if } i \in \mathcal{T} \end{cases}$ *. Let Eq.(19) be a mapping between the log-potential* $q_i$ *(*$\widehat{q}_i$*) and potential* $u_i$ *(*$\widehat{u}_i$*) for* $i \in \mathcal{V}$ *(*$i \in \widehat{\mathcal{V}}$*).*

$$q_i^t = \log\left((1-u_i^t)(1-\delta) + \delta\right), \tag{19}$$

*where* $0 < \delta < 1$ *denotes a small constant. Then, mapping a Richardson update (Eq.(17)) in original space by Eq.(19) is equivalent to a Richardson update Eq.(18) in the log-space.*

**Remarks:** Eq.(18) is a Log-Mean-Exp nonlinear aggregation, it enables us to update the potential in the log space instead of the original potential space. In practice, only $q_i^t$ for $i \in \{1, \cdots, n\}$ is updated in the log-space. We do not need to compute the potential $u_i^t$ in the original potential space explicitly by taking advantage of Log-Sum-Exp trick. This updating scheme can reduce the numeric issue of *float* reported in (Wray et al., 2016).

With the update rule in Eq.(18), our GPFNN maintains the optimization property. Namely, it performs a $T$-step Richardson update in the log-space. It is worth to remark that this nonlinear update can reduce the numeric error compared with a linear update in original potential space. Thus, our nonlinear update can maintain significantly smaller accumulation numeric error compared linear update, especially when $T$ is large.

## 4 Graph Learning Framework

In this section, we present a simple architecture in GPFNN family for graph learning. We fix the weight matrix $\boldsymbol{W}_1^t = \boldsymbol{W}_2^t = \boldsymbol{I}$ for all the $T$-layers, and apply the non-linear aggregation in Eq.(18) with adaptive stepsizes $\boldsymbol{\alpha} = [\alpha_1, \cdots, \alpha_n]^\top$ that automatically computed by Eq.(16). Remarkably, our adaptive stepsize scheme is more flexible to update each node. Moreover, it greatly saves the human efforts regarding fine-tuning the hyperparameter $[\alpha_1, \cdots, \alpha_n]$ for each node.

For graph learning problems, we learn the parameters of the initial potential neural network and the visual node potential neural network. Details are shown in the following subsections.

### 4.1 Simple Structure of GPFNN for Graph Learning

#### 4.1.1 Initial Potential Neural Network

The initial potential of the nodes ($\boldsymbol{U}^0$) can be learned using a neural network. The architecture of the Initial Potential Neural Network (IPNN) can either be a simple linear mapping, or a complex neural network depends on the tasks.

To compatible with the potential update Eq.(18) in log space, we employ the Log-Sigmoid function as the activation function of the last layer of IPNN:

$$\sigma(z) = \log\left(\frac{1}{1 + \exp(-z)}\right) = z - \log\left(1 + \exp(z)\right). \tag{20}$$

In this work, we employ a three layer architecture as our IPNN given below:

$$\boldsymbol{U}^0 = \sigma\left(\boldsymbol{Z}\boldsymbol{W}_a\right), \tag{21}$$

where $\boldsymbol{Z} = [\boldsymbol{Z}_1, \cdots, \boldsymbol{Z}_m]$ denotes a concatenation of $m$ blocks. The $i^{th}$ block $\boldsymbol{Z}_i$ is given as follows:

$$\boldsymbol{Y}_i^{(1)} = Dropout(\boldsymbol{X}^{2i-1}) \tag{22}$$

$$\boldsymbol{Y}_i^{(2)} = ReLU(\boldsymbol{Y}_i^{(1)}\boldsymbol{W}_{a1}^i) + \boldsymbol{Y}_{i-1}^{(2)} - Mean(\boldsymbol{Y}_{i-1}^{(2)}) \tag{23}$$

$$\boldsymbol{Z}_i = LayerNorm(\boldsymbol{Y}_i^{(2)}), \tag{24}$$

where $\boldsymbol{X}^{2i-1}$ denotes the input feature of nodes after $(2i-1)$-hop message passing. $\boldsymbol{X}^0$ denotes the initial input node features. The $ReLU(\cdot)$, $Dropout(\cdot)$, $Mean(\cdot)$, and $LayerNorm(\cdot)$ denote the ReLU activation function, dropout, mean function and layer normalization function respectively. In addition, we set $\boldsymbol{Y}_0^{(2)} = LayerNorm(ReLU(\boldsymbol{Y}_0^{(1)}\boldsymbol{W}_{a1}^i))$.

#### 4.1.2 Visual Node Potential Neural Network

The potential of the visual nodes ($\widehat{\boldsymbol{U}}$) can also be learned using a neural network. The architecture of the Visual Node Potential Neural Network (VNPNN) can either be same as IPNN, or a more complex neural network as a teacher model.

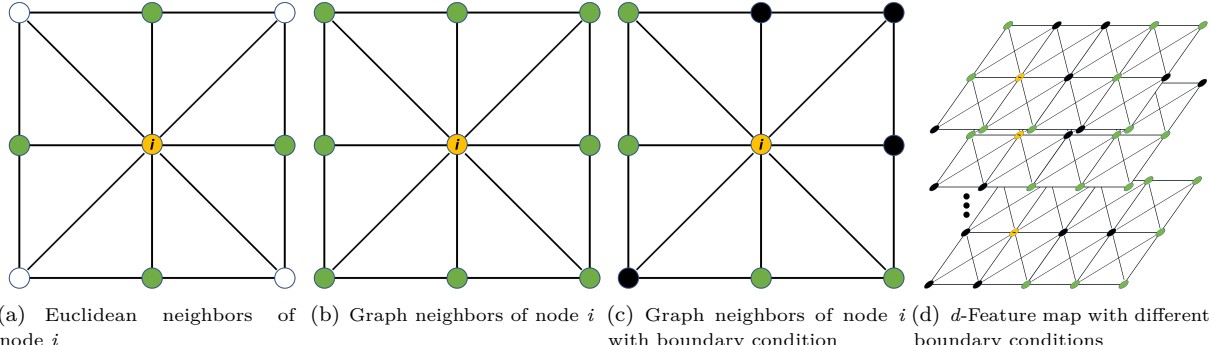

(a) Euclidean neighbors of node $i$    (b) Graph neighbors of node $i$    (c) Graph neighbors of node $i$ with boundary condition    (d) $d$-Feature map with different boundary conditions

Figure 2: Illustration of the neighbors of node $i$ in the Euclidean topology and the graph topology, and a $d$-dimensional feature map with different boundary condition on each dimension.

To compatible with the potential update Eq.(18) in log space, we employ the Log-Sigmoid function as the activation function of the last layer of VNPNN:

$$\sigma(z) = \log\left(\frac{1}{1 + \exp(-z)}\right). \tag{25}$$

In this work, we employ a three layer architecture as our VNPNN given below

$$\widehat{\boldsymbol{U}} = \sigma\left(\bar{\boldsymbol{Z}}\boldsymbol{W}_b\right), \tag{26}$$

where $\bar{\boldsymbol{Z}} = [\bar{\boldsymbol{Z}}_1, \cdots, \bar{\boldsymbol{Z}}_m]$ denotes a concatenation of $m$ blocks. The $i^{th}$ block $\bar{\boldsymbol{Z}}_i$ is given as follows:

$$\bar{\boldsymbol{Y}}_i^{(1)} = Dropout(\boldsymbol{X}^{2i-2}) \tag{27}$$

$$\bar{\boldsymbol{Y}}_i^{(2)} = ReLU(\bar{\boldsymbol{Y}}_i^{(1)}\boldsymbol{W}_{b1}^i) + \bar{\boldsymbol{Y}}_{i-1}^{(2)} - Mean(\bar{\boldsymbol{Y}}_{i-1}^{(2)}) \tag{28}$$

$$\bar{\boldsymbol{Z}}_i = LayerNorm(\bar{\boldsymbol{Y}}_i^{(2)}), \tag{29}$$

where $\boldsymbol{X}^{2i-2}$ denotes the input feature of nodes after $(2i-2)$-hop message passing. $\boldsymbol{X}^0$ denotes the initial input node features. In addition, we set $\bar{\boldsymbol{Y}}_0^{(2)} = LayerNorm(ReLU(\bar{\boldsymbol{Y}}_0^{(1)}\boldsymbol{W}_{b1}^i))$.

### 4.2 Contrastive Regularization between Visual Node Potential and Initial Potential

We provide a regularization technique by introducing a regularization loss between the Visual Node Potential and Initial Potential. The contrastive loss is given as follows:

$$\ell_c(\boldsymbol{U}^0, \widehat{\boldsymbol{U}}) = \|\boldsymbol{U}^0 - \widehat{\boldsymbol{U}}\|_F^2. \tag{30}$$

The contrastive loss $\ell_c$ regularize the $Frobenius$-norm between the initial log-potential and the Visual Node log-potential. It forces the initial log-potential close to the Visual Node log-potential, which provides a good prior initial log-potential that leads to fast converge (for graph Richardson update). This is a nice protery that enables us to use a small number of (Richardson update) layers to achieve a near stationary global potential field.

Moreover, when the input feature representation of the same nodes for the IPNN and VNPNN are different as section 4.1, our loss can be viewed as a contrastive learning scheme. This provides a way to learn useful representations of nodes that are semantically meaningful and invariant to certain transformations of the input graph.

## 5 Fast GPFNN on Grid Worlds

In this section, we show how to apply our GPFNN on grid worlds, and achieve a neat parallel local update for fast computation. The 2D-grid worlds relate to a sparse graphs with maximum degree no larger than

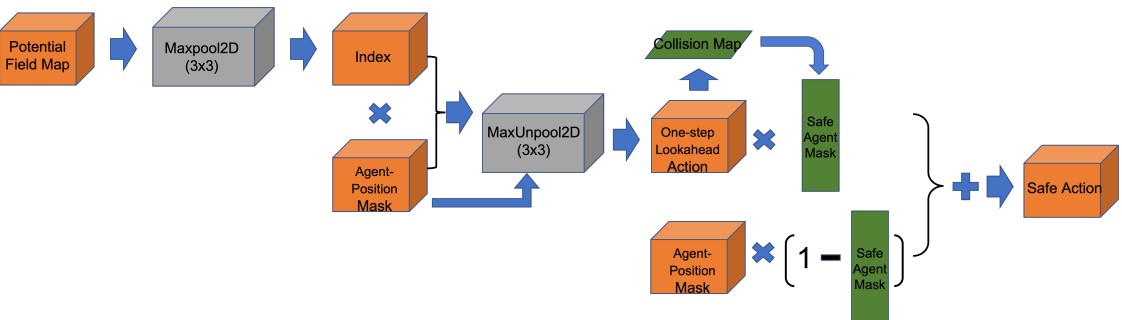

Figure 3: Illustration of Network Architecture of Action Decoder

eight. See Figure 1 for an illustration. We can see that the maze in Fig 1(a) has an equivalent graph in Fig 1(c). Thus, computing a valid potential field on the maze (Fig 1(b)) leads to a valid potential field on graph in Fig 1(b) (with boundary condition on virtual nodes). The walls and targets on mazes plays a role of virtual nodes of the embedded equivalent graph.

## 5.1 Neat parallel local update for fast computation on grids

For a 2D-grid world, the cost weight $c_{ij} = 1$. We set the hyper-parameter $\alpha = \frac{8}{9}$. This leads to the following neat parallel local update rule similar to a masked 2D average pooling (with a $3 \times 3$ kernel size) :

$$u_i^{t+1} = \frac{u_i^t + \sum_{j \in \mathcal{N}(i)} \mathbf{1}(j \in \mathcal{V})u_j^t + \mathbf{1}(j \in \widehat{\mathcal{V}})\widehat{u}_j}{9}, \tag{31}$$

for all $i \in \mathcal{V}$, where $\mathcal{V}$ denotes the node-set with free update potential parameter, $\widehat{\mathcal{V}}$ denotes the node-set of boundary condition, and $\mathbf{1}(\cdot)$ denote indicator function. See an illustration in Fig 2(c). The green nodes denote the free updated nodes that belong to node-set $\mathcal{V}$, and the black nodes denote the boundary nodes that belongs to the node-set $\widehat{\mathcal{V}}$.

When we employ the Log-Mean-Exp aggregation, i.e., $\sigma(\boldsymbol{x}) = \log \frac{1}{9} \sum_k \exp{(x_k)}$ instead of an average map, we have update rule in a log-space for all $i \in \mathcal{V}$ as Eq.(32).

$$q_i^{t+1} = \log\Big(\frac{\exp(q_i^t) + \sum_{j \in \mathcal{N}(i)} \exp{\big(\mathbf{1}(j \in \mathcal{V})q_j^t + \mathbf{1}(j \in \widehat{\mathcal{V}})\widehat{q}_j\big)}}{9}\Big). \tag{32}$$

The update rule in Eq.(32) can be implemented by Conv2D and Log-Sum-Exp that are fast and stable in modern deep learning toolboxes.

## 5.2 Cooperative actions for $d$ agents
To take cooperative actions for $d$ agents, we construct $d$ maps, one for each agent. The walls and target sets are the boundary conditions same for all agents. Further, for each agent $i$, we add other agents' position as the obstacle nodes to boundary conditions of the $i^{th}$ map. This results in a $d \times w \times h$ sized 3D-tensor, where $w, h$ denotes the width and height of the grid map. See Figure 2(d) for an illustration.

We employ the $T$-step update in Eq.(32) as a $T$-layer feedforward neural network, it outputs the feature map $\boldsymbol{U}^T$ as a $d \times w \times h$ sized 3D-tensor, where $w, h$ denotes the width and height of the grid map. The feature map $\boldsymbol{U}^T$ is then fed into an action decoder network compute each agent's action in parallel. The action decoder network computes one-step lookahead of all the agents to avoid a collision. The unsafe lookahead action will be rollback.

Our model can leverage the user preference of the priority of each agents. When several agents take action to the same position, the agent with the highest priority (largest ID in default) will indeed take action to move, and the others' lookahead action will be rollback and stay hold on.

An illustration of the architecture of the action decoder network is shown in Figure 3. The agent-position-mask is the one-hot representation of the agents' current positions. The action decoder computes the collision map based on agents' one-step lookahead action. Then, it constructs a safe-agent-mask. Agents with safe-agent-mask one perform the lookahead action, while agents with safe-agent-mask zero remain in the old position. The action networks consists of a MaxPooling2D layer, a MaxUnPool2D layer. The MaxPooling2D layer enables to compute the maximum potential in a parallel way. The MaxUnPool2D layer is used to find the one-step lookahead position for each agent in parallel.

It is worth to noting that our GPFNN never lead to edge conflict (two agents came across the same edge). This is because each agent treat other agents' as obstacle nodes, its original position always has a lower potential than the obstacle nodes. Thus, the agent will stay at the original node instead of moving to other agents along with the directly connected edge. This avoids the edge conflict.

## 6 Experiments

### 6.1 Continual Group-wise Multi-agent Path Planning for Static Targets

To evaluate the planning ability of our GPFNN, we first perform empirical studies on group-wise multi-agent path planning tasks with static targets. More specifically, in our experiments, a group of agents navigate on the map to catch a group of static targets. The agents are moving in parallel during each time step. The agents perform as a team to catch as many targets as possible while using as few time steps as possible. We evaluate our GPFNN on complex 2D grid mazes. We create three different-sized 2D mazes for evaluation, i.e., $129 \times 129$-sized mazes, $401 \times 401$-sized mazes, and $1025 \times 1025$-sized mazes. Each size contains 100 mazes.

We run one trial for each maze among the 100 same-sized mazes with randomly generated non-overlapped targets and agents' initialized positions. For $129 \times 129$-sized mazes, and $401 \times 401$-sized mazes, we test four cases with different numbers of agents, namely, 50 agents cases, 100 agents cases, 500 agents cases, and 1000 agents cases. For the large $1025 \times 1025$-sized mazes, we test 50 agents and 100 agents cases. The number of static targets is the same as the number of agents.

We compare GPFNN with related methods: Conflict-Based Search (Sharon et al., 2015), Enhanced Conflict-Based Search (ECBS) (Barer et al., 2014) and Prioritized Planning (P-Planning) (Čáp et al., 2015). For all the baselines, we use the C++ codes in the Multi Robot Planning toolbox [1]. For all baselines, we employ random agent-target pair assignment. In all the experiments, we set the number of layers of GPFNN to 1000. The maximum moving step for GPFNN is set to 2000. The "Time out" parameter of CBS is set to $72,000$ seconds.

We employ three metric to measure the effectiveness of different methods. The metric employed are listed below:

- **Scores:** The score of each trail is defined as the number of targets been caught in the trail. We report the mean scores $\pm$ Standard Deviation over all trails for evaluation.

- **Success Rate:** The success rate is defined as the percentage of the success trails over the total trails, i.e., $SR := \frac{Number\ of\ Success\ Trails}{Total\ Number\ of\ Trails}$. We said a success trail when all the targets in this trail have been caught.

- **Finishing Time Steps:** The finishing time step of a target is the time step that the target has been caught. We report the mean Finishing Time Step $\pm$ Standard Deviation over all targets caught in all trails for evaluation.

The mean scores $\pm$ std of different methods for 50 agents, 100 agents, 500 agents and 1000 agents cases on different sized mazes are shown in Table 1. We can observe that GPFNN attains near the maximum scores for each number of agent cases. It significantly outperforms CBS, ECBS and P-Planning regarding the scores. Moreover, we can see that CBS, ECBS and P-Planning obtain large scores on the small mazes,

---

[1] https://github.com/whoenig/libMultiRobotPlanning

Table 1: Mean Score $\pm$ std for different number of agents on different maps (larger is better)

| | Number of Agents | GPFNN | CBS | ECBS | P-Planning |
|---|---|---|---|---|---|
| 129x129-maze | 50 | **50 $\pm$ 0.00** | 0 | 35.55 $\pm$ 3.46 | 37.46 $\pm$ 3.54 |
| | 100 | **100 $\pm$ 0.00** | 0 | 70.63 $\pm$ 5.21 | 73.83 $\pm$ 5.96 |
| | 500 | **500 $\pm$ 0.00** | 0 | 0 | 377.72 $\pm$ 17.43 |
| | 1000 | **1000 $\pm$ 0.00** | 0 | 0 | 845.11 $\pm$ 26.76 |
| 401x401-maze | 50 | **50 $\pm$ 0.00** | 0 | 5.25 $\pm$ 1.56 | 4.90 $\pm$ 1.79 |
| | 100 | **100 $\pm$ 0.00** | 0 | 0 | 9.56 $\pm$ 2.88 |
| | 500 | **499.92 $\pm$ 0.80** | 0 | 0 | 55.89 $\pm$ 7.13 |
| | 1000 | **1000 $\pm$ 0.00** | 0 | 0 | 27.19 $\pm$ 6.12 |
| 1025x1025-maze | 50 | **49.92 $\pm$ 0.27** | 0 | 0 | 7.68 $\pm$ 2.79 |
| | 100 | **99.50 $\pm$ 4.97** | 0 | 0 | 15.21 $\pm$ 3.49 |

while they only achieve very small scores on the challenging large-sized mazes, i.e., $401 \times 401$ sized mazes and $1025 \times 1025$ sized mazes. This shows that these baselines cannot scale up to large-sized maps.

More experiment results on continual group-wise path planning for static targets and dynamic targets can be found in section C and section D in the Appendix, respectively. Furthermore, we evaluate GPFNN in the TAPF setting. The experimental results are shown in section E. Moreover, we further evaluate the learning ability of our GPFNN in section F in the Appendix. The results on six graph datasets (up to millions of nodes) demonstrate that our GPFFN achieves competitive node classification accuracy with a very small number of parameters.

## 7 Limitation

Currently, our work focuses on path planning for homogeneous agents. Path planning for multiple heterogeneous agents presents a more challenging problem that needs further investigation. One potential approach is to adopt a hierarchical planning strategy. At the high level, our GPFNN could be utilized to perform abstract planning in a homogeneous way, while at the low-level, high-resolution grids could be employed to perform a more precise, high-resolution planning tailored to the specific characteristics of the heterogeneous agents. Another limitation is that our GPFNN can not perform real-time applications that require performing path planning in seconds.

## 8 Conclusion and Future Work

This paper proposed a graph potential field-based neural network (GPFNN) for the group-wise homogeneous multi-agent path planning. Our GPFNN unfolds the optimization steps of the potential field as a T-layer feedforward neural network. Theoretically, our GPFNN has a linear convergence rate to a valid potential field map w.r.t. the depth despite learning the diffusion weights. Empirically, our GPFNN achieves superior performance for up to 1000 agents group-wise path planning on $129 \times 129$-sized and $401 \times 401$-sized complex mazes and up to 100 agents on the large-sized $1025 \times 1025$ mazes. Moreover, our GPFNN supports catching dynamic moving, even adversarial moving targets for 500 agents' cooperative planning on $129 \times 129$-sized and $401 \times 401$-sized complex mazes, and 50 agents' cooperative planning on the large-sized $1025 \times 1025$ mazes. The experiments on planning tasks demonstrate our GPFNN's strong group-wise path-planning ability.

In addition, we empirically evaluated the learning ability of our GPFNN on graph node classification tasks. The experiments on six graph datasets demonstrate the learning ability of our GPFNN with a small number of parameters. This may show the potential of our GPFFN to handle the tasks that require both planning ability and learning ability, e.g., multi-task learning that supports multi-agent path planning and node action prediction (node classification). Moreover, GPFNN may be a useful component in multi-agent reinforcement learning in tasks requiring path planning and team-wise cooperative actions, e.g., multi-agent RL in city package delivery, multi-agent RTS/MOBA games, etc. We leave these interesting applications as one of our future works.

## Acknowledgments

This work is supported by Career Development Fund (CDF) of the Agency for Science, Technology and Research (A*STAR) (No: C243512014).

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

# Appendix

## A   Proof of Proposition 1

**Proposition.** *Let the boundary condition of potential in original space be* $\widehat{u}_i = \begin{cases} 1 & \text{if } i \in \mathcal{O} \\ 0 & \text{if } i \in \mathcal{T} \end{cases}$ *. Then, mapping a Richardson update (Eq.(17)) in original space by Eq.(19) is equivalent to a Richardson update Eq.(18) in the log-space.*

*Proof.* We first check the boundary condition. When $i \in \mathcal{O}$, from the mapping Eq.(19), we know that

$$
\begin{aligned}
\widehat{q}_i &= \log\left((1 - \widehat{u}_i)(1 - \delta) + \delta\right) \\
&= \log\left((1 - 1)(1 - \delta) + \delta\right) = \log(\delta)
\end{aligned}
\tag{33}
$$

When $i \in \mathcal{T}$, we know that

$$
\begin{aligned}
\widehat{q}_i &= \log\left((1 - \widehat{u}_i)(1 - \delta) + \delta\right) \\
&= \log\left((1 - 0)(1 - \delta) + \delta\right) = \log(1) = 0
\end{aligned}
\tag{34}
$$

When $i$ belongs to the free node set, i.e., $i \in \mathcal{V}$, from the mapping Eq.(19) and update rule Eq.(17), we know that

$$
q_i^{t+1} = \log\left((1 - u_i^{t+1})(1 - \delta) + \delta\right)
$$

$$
= \log\left(\left(1 - \frac{u_i^t + \sum_{j \in \mathcal{N}(i)} u_j^t + \widehat{u}_i}{d_i + 2}\right)(1 - \delta) + \delta\right)
$$

$$
= \log\left(\left(\frac{1}{d_i + 2} + \frac{d_i + 1}{d_i + 2} - \frac{u_i^t + \sum_{j \in \mathcal{N}(i)} u_j^t + \widehat{u}_i}{d_i + 2}\right)(1 - \delta) + \delta\right)
\tag{35}
$$

$$
= \log\left(\left(\frac{1 - u_i^t}{d_i + 2} + \frac{\sum_{j \in \mathcal{N}(i)}\left(1 - u_j^t\right) + 1 - \widehat{u}_i}{d_i + 2}\right)(1 - \delta) + \delta\right)
\tag{36}
$$

$$
= \log\left(\frac{(1 - u_i^t)(1 - \delta)}{d_i + 2} + \frac{\sum_{j \in \mathcal{N}(i)}\left(1 - u_j^t\right)(1 - \delta) + (1 - \widehat{u}_i)(1 - \delta)}{d_i + 2} + \delta\right)
\tag{37}
$$

$$
= \log\left(\frac{(1 - u_i^t)(1 - \delta) + \delta}{d_i + 2} + \frac{\sum_{j \in \mathcal{N}(i)}\left((1 - u_i^t)(1 - \delta) + \delta\right) + (1 - \widehat{u}_i)(1 - \delta) + \delta}{d_i + 2}\right)
\tag{38}
$$

Note that $\exp\left(q_j^t\right) = (1 - u_j^t)(1 - \delta) + \delta$ for $j \in \mathcal{N}(i)$, and $\exp\left(q_i^t\right) = (1 - u_i^t)(1 - \delta) + \delta$, and $\widehat{q}_i = \log\left((1 - \widehat{u}_i)(1 - \delta) + \delta\right)$, it follows that

$$
q_i^{t+1} = \log\left(\frac{\exp\left(q_i^t\right)}{d_i + 2} + \frac{\sum_{j \in \mathcal{N}(i)} \exp\left(q_j^t\right) + \exp\left(\widehat{q}_i\right)}{d_i + 2}\right)
\tag{39}
$$

$$
= \log\left(\frac{\exp\left(q_i^t\right) + \sum_{j \in \mathcal{N}(i)} \exp\left(q_j^t\right) + \exp\left(\widehat{q}_i\right)}{d_i + 2}\right)
\tag{40}
$$

$\square$

## B   Demonstration of Potential Filed Constructed by GPFNN

We first show a demonstration of the potential field constructed by GPFNN in Figure 4. The number of layers of GPFNN is set to 1000 to construct the potential field. From both the Figure 4(a) and Figure 4(b), we can observe that the brighter the color is, the closer to the exit in the right-down corner. which shows that the potential field provides an monotonic guide from any position to the exit (if connected). This is vitally important for multi-agent path planning to construct cooperatively moving action toward the targets.

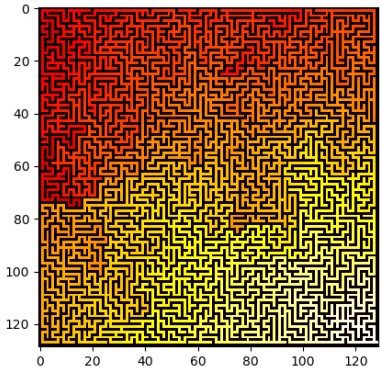

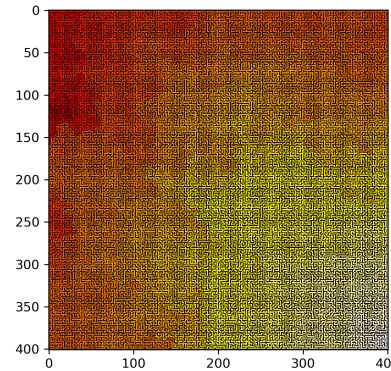

(a) Potential Field Map on a 129 × 129 sized Maze

(b) Potential Field Map on a 401 × 401 sized Maze

Figure 4: Demonstration of the potential field maps constructed by our GPFNN on a 129 × 129-sized maze and a 401 × 401-sized maze. The brighter the color is, the closer to the exit.

## C Continual Group-wise Multi-agent Path Planning for Static Targets

The success rate of 50 agents, 100 agents, 500 agents, and 1000 agents cases on different sized mazes are shown in Table 2. We can observe that CBS, ECBS and Prioritized Planning fail to catch all the targets in any trail. This is because these methods are designed for one-to-one agent-target pair path planning. They cannot perform group-wise multi-agent path planning as a team. The planned path for each individual agent may always have a conflict for complex and dense maps. As a result, they all fail to catch all targets in any trial on complex mazes with dense obstacles. In contrast, our GPFNN performs group-wise multi-agent path planning as a team. Our GPFNN implicitly assigns the target to the nearby agent according to the potential field. Thus, our GPFNN can handle path planning with a large number of agents, while other baselines cannot.

The mean finishing time steps ± std for 50 agents, 100 agents, 500 agents and 1000 agents cases on different sized mazes are shown in Table 3. We can observe that GPFNN achieves smaller and smaller mean finishing time steps as the number of agents increases. This demonstrates the ability of GPFNN to plan multi-agent paths as a whole team instead of individually. Thus, when the number of agents in a team gets larger, GPFNN can find better path planning that catches the targets with fewer steps. In contrast, the baselines CBS, ECBS and P-Planning all perform the multi-agent path planning for the one-to-one agent-target pairs individually instead of cooperating as a team. They only perform planning to avoid a collision but fail to cooperate as a team to catch the group-wise targets. As a result, they perform worse as the number of agents increases for group-wise multi-agent path planning. This can be observed in Table 1 and Table 2, in which, CBS failed to output a path for all cases inside the time limitation (72,000 seconds). In addition, both CBS and ECBS fail to output a path for more than 100 agents on all three sized mazes within the time limitation (72,000 seconds).

In addition, we can see that P-Planning obtains smaller mean finishing time steps from Table 3. Note that P-Planning has a zero Success Rate to catch all the targets in each trial from Table 2 and achieves small scores from Table 1. This shows that P-Planning may prioritize catching the easy targets (the closed targets) and fail to catch the challenging ones (far away targets). As a result, the mean finishing time steps over the targets successfully caught are small.

## D Continual Group-wise Multi-agent Path Planning for Dynamic Targets

In this section, we employ our GPFNN to address the multi-agent group-wise dynamic path planning on 2D mazes. This task is very challenging because it requires planning as a team instead of achieving individual

Table 2: Success Rate for Different Number of Agents on Different Maps

|  |  | GPFNN | CBS | ECBS | P-Planning |
|---|---|---|---|---|---|
| 129x129-maze | 50 | 100% | 0 | 0 | 0 |
|  | 100 | 100% | 0 | 0 | 0 |
|  | 500 | 100% | 0 | 0 | 0 |
|  | 1000 | 100% | 0 | 0 | 0 |
| 401x401-maze | 50 | 100% | 0 | 0 | 0 |
|  | 100 | 100% | 0 | 0 | 0 |
|  | 500 | 99% | 0 | 0 | 0 |
|  | 1000 | 100% | 0 | 0 | 0 |
| 1025x1025-maze | 50 | 92% | 0 | 0 | 0 |
|  | 100 | 99% | 0 | 0 | 0 |

Table 3:   Mean Finishing Time Steps $\pm$ Std for different number of Agents on different maps (Smaller is better)

|  |  | GPFNN | CBS | ECBS | P-Planning |
|---|---|---|---|---|---|
| 129x129-maze | 50 | $22.43 \pm 2.50$ | - | $116.18 \pm 79.84$ | $122.47 \pm 46.80$ |
|  | 100 | $15.44 \pm 1.28$ | - | $128.21 \pm 78.11$ | $121.22 \pm 43.63$ |
|  | 500 | $5.65 \pm 0.27$ | - | - | $72.43 \pm 68.97$ |
|  | 1000 | $3.59 \pm 0.10$ | - | - | $36.0 \pm 55.73$ |
| 401x401-maze | 50 | $118.89 \pm 17.56$ | - | $59.5 \pm 107.33$ | $24.36 \pm 69.07$ |
|  | 100 | $82.66 \pm 7.73$ | - | - | $11.67 \pm 50.56$ |
|  | 500 | $35.35 \pm 37.76$ | - | - | $14.87 \pm 53.67$ |
|  | 1000 | $20.08 \pm 0.79$ | - | - | $9.07 \pm 43.22$ |
| 1025x1025-maze | 50 | $328.46 \pm 32.07$ | - | - | $122.68 \pm 269.53$ |
|  | 100 | $243.25 \pm 17.51$ | - | - | $46.16 \pm 180.13$ |

goals. More importantly, the targets are dynamically moving, even adversarially moving, on the 2D mazes with dense obstacles.

### D.1   2D-Grid World Problems Setup

In the experiments, $N$ agents are assigned to groups $A$ and $B$, respectively. Both groups $A$ and $B$ agents are dynamically moving on the maze. The mission of group $A$ is to catch the agents in group $B$. Once an agent in group $A$ gets in the same position as an agent in group $B$, group $A$'s score plus one, and the agent of group $B$ in this position is removed. The mission of group $B$ is to catch $N$ static targets. The static targets of group $B$ do not move. Once an agent of group $B$ catches a static target, group $B$'s score plus one, and the target is removed. The group that first attains a $N$ score wins the game.

Both $129 \times 129$-sized mazes, $401 \times 401$-sized mazes, and the large $1025 \times 1025$-sized mazes are employed for evaluation. For $129 \times 129$-sized mazes and $401 \times 401$-sized mazes, we employ 100 mazes for evaluation, for the large $1025 \times 1025$-sized mazes, we employ 50 mazes for evaluation. For experiments on $129 \times 129$-sized mazes, $401 \times 401$-sized mazes, we set the number of agents(targets) to 500, i.e., $N = 500$. For experiments on the large $1025 \times 1025$-sized mazes, we set $N = 50$. For group $A$, we employ the policy generated by our GPFNN. For group $B$, three kinds of policy are employed for comparison: *(1)* adversarial policy using GPFNN planning to targets with group $A$'s position information, *(2)* unaware policy using GPFNN planning to targets without group $A$'s position information, *(3)* random moving policy with one-step look-ahead collision checking (stay hold on if collision will happen).

At each round (time step), group $A$'s agents first move in parallel, then group $B$'s agents move in parallel. The initialed position of group $A$'s agents, group $B$'s agents, and the static targets of group $B$ are randomly assigned without overlap and collision.

Table 4: Mean Scores $\pm$ std for Group A (GPFNN) and Group B with different Policy

| | GPFNN vs Random Group B | | GPFNN vs Unaware Group B | | GPFNN vs Adversarial Group B | |
|---|---|---|---|---|---|---|
| | Group A's Score | Group B's Score | Group A's Score | Group B's Score | Group A's Score | Group B's Score |
| 129x129-maze | $500 \pm 0$ | $34.13 \pm 6.21$ | $500 \pm 0$ | $237.13 \pm 11.60$ | $500 \pm 0$ | $251.37 \pm 13.49$ |
| 401x401-maze | $500 \pm 0$ | $14.31 \pm 4.12$ | $500 \pm 0$ | $258.49 \pm 11.40$ | $500 \pm 0$ | $311.04 \pm 33.59$ |
| 1025x1025-maze | $50 \pm 0$ | $0.08 \pm 0.27$ | $50 \pm 0$ | $36.30 \pm 4.73$ | $49.00 \pm 3.58$ | $47.04 \pm 2.12$ |

Table 5: Mean Finishing Time Steps $\pm$ std of Group A (GPFNN) against Group B with different Policy

| | GPFNN vs Random Group B | GPFNN vs Unaware Group B | GPFNN vs Adversarial Group B |
|---|---|---|---|
| 129x129-maze | $37.12 \pm 10.28$ | $22.25 \pm 3.79$ | $31.75 \pm 6.07$ |
| 401x401-maze | $180.10 \pm 39.97$ | $125.03 \pm 19.44$ | $191.54 \pm 40.11$ |
| 1025x1025-maze | $1057.68 \pm 227.93$ | $1191.74 \pm 208.76$ | $1704.12 \pm 429.23$ |

## D.2 Experimental Results for Group-wise Multi-agent Path Planning with Dynamic Targets

The mean scores of Group A against Group B with different policies are shown in Table 4. From Table 4, we can observe that Group A with GPFNN planning attains the full scores on all the cases except the case against the Adversarial Group B on $1025 \times 1025$-sized mazes, which shows the power of GPFNN perform the duty of Group A. Note that Group A has an advantage over Group B because of Group A can remove Group B's agents, but Group B can not remove Group A's agents. This advantage is more significant for the dense agents' cases on $129 \times 129$-sized mazes and $401 \times 401$-sized mazes.

Moreover, we can observe that Group B with GPFNN planning (Unaware Group B and Adversarial Group B) achieves significantly higher scores compared with Random Group B, which shows the importance of planning. Moreover, as the maze size increases, the score of random policy decreases. Particularly, Random Group B attained almost zero scores on the large $1025 \times 1025$-sized mazes, which shows that random walk cannot perform well on large-sized mazes. In addition, the Adversarial Group B attains higher scores on all-sized mazes than Unaware Group B, which shows the ability of GPFNN to take advantage of Group A's position information to increase the scores.

The mean finishing time of Group A catching all $N$ targets against Group B with different policies is shown in Table 5. From Table 5, we can see that the fishing time step increases as the size of the maze increases. Moreover, the Adversarial Group B cases attain larger mean finishing steps compared with Unaware Group B, especially on the large $1025 \times 1025$-sized mazes, which shows that GPFNN's adversarial planning ability (with Group A's position information) to increase the survival time of Group B.

We provide the GIF video demonstrations of GPFNN Group A against Adversarial GPFNN Group B in the supplementary materials. In the demonstrations, each red point denotes each Group A's agent, each blue point denotes each Group B's agent, and each green point denotes each static target of Group B. In the demo of 10 agents on the 129x129-sized maze, group B (blue) first catches all the static targets (green) and wins the game. In the demo of 20 agents, 50 agents, and 100 agents case on the 129x129-sized maze, group A (red) first catches all Group B's agents (blue) and wins the game. In the demo of 500 agents on the 401x401-sized maze, group A (red) first catches all Group B's agents (blue) and wins the game. From these demos, we can see that for the dense agents' cases, Group A has more advantage than Group B because Group A can remove Group B's agents during the game process, but Group B can not remove Group A's agents. For the sparse agents' case, Group B with GPFNN still has a chance to win Group A with GPFNN, e.g., 10 agents case. More interestingly, we can observe Group A's team behavior in that agents cooperatively besiege Group B's agents from different roads(directions) on the mazes instead of performing catching individually as a queue.

## E Evaluation planning ability in TAPF setting

We further evaluate the planning ability of our GPFNN in the TAPF setting. Compared with our continual group-wise planning problems studied in section C and section D, the TAPF setting requires the agent to stop once it reaches the target. In contrast, in our continual group-wise planning problems, the agents that achieve (reach) the target can continue to search for the remaining targets.

Table 6: Success Rate for Different Number of Agents on Different Maps(larger is better)

| | | GPFNN | CBS-TA | ECBS-TA |
|---|---|---|---|---|
| 54x54-maze | 10 | 100% | 97% | 97% |
| | 50 | 100% | 3% | 87% |
| | 100 | 97% | 0 | 76% |
| | 200 | 74% | 0 | 57% |
| 250x250-maze | 10 | 100% | 0 | 0 |
| | 50 | 100% | 0 | 0 |
| | 100 | 100% | 0 | 0 |
| | 200 | 100% | 0 | 0 |

Table 7: Mean Finishing Time Steps $\pm$ Std for different number of Agents on different maps (Smaller is better)

| | | GPFNN | CBS-TA | ECBS-TA |
|---|---|---|---|---|
| 54x54-maze | 10 | $178.89 \pm 62.56$ | $70.89 \pm 11.07$ | $70.95 \pm 11.02$ |
| | 50 | $233.12 \pm 50.62$ | $91.01 \pm 4.32$ | $83.63 \pm 7.63$ |
| | 100 | $239.33 \pm 60.30$ | - | $87.94 \pm 6.47$ |
| | 200 | $274.35 \pm 74.01$ | - | $94.87 \pm 6.77$ |
| 250x250-maze | 10 | $289.66 \pm 84.48$ | - | - |
| | 50 | $411.36 \pm 89.12$ | - | - |
| | 100 | $458.43 \pm 98.40$ | - | - |
| | 200 | $502.65 \pm 103.76$ | - | - |

We compare GPFNN with Conflict-Based Search with Optimal Task Assignment (CBS-TA) (Hoenig et al., 2018) and Enhanced Conflict-Based Search with Optimal Task Assignment (ECBS-TA). For all the baselines, we use the C++ codes in the Multi Robot Planning toolbox [2]. In all the experiments, we set the number of layers of GPFNN to 1000. The maximum moving step for GPFNN is set to 3000. The "Time out" parameter of CBS-TA and ECBS-TA is set to $72,000$ seconds.

We evaluate all methods on $54 \times 54$ sized maze and $250 \times 250$ sized maze. The mazes used here have wider roads compared with the mazes in our continual group-wise planning setting. For each maze, we test four cases with different numbers of agents, namely, 10 agents cases, 50 agents cases, 100 agents cases, and 200 agents cases. The number of targets is set to the same number as that of the agents. For each case, we perform 100 independent trials with randomly initialized agents' and targets' positions.

The success rate for different numbers of agents on different maps are reported in Table 6. We can observe that GPFNN achieve a consistently higher success rate compared with CBS-TA and ECBS-TA. In addition, we can see that in the 10-agent case on the $54 \times 54$ sized maze, both GPFNN and baselines achieve a high success rate. It shows that all the methods can perform well on the small mazes with a small number of agents. Moreover, the success rate of CBS-TA and ECBS-TA decrease fast as the number of agents increases. More importantly, CBS-TA and ECBS-TA obtain zero success rates on the $250 \times 250$ sized maze, while GPFNN achieves 100% success rate. This demonstrates the planning ability of GPFNN on large maps compared with CBS-TA and ECBS-TA.

The mean finishing time steps for different numbers of agents on different maps are shown in Table 7. Because the CBS-TA and ECBS-TA only output transitory files for the success cases, i.e., all targets have been caught. The mean finishing time steps for both GPFNN and baselines are computed over all success trials. From Table 7, we see that CBS-TA and ECBS-TA achieve smaller mean finishing time steps compared with GPFNN on the small maze. This may show that CBS-TA and ECBS-TA can find better paths on the small mazes. Moreover, we can see that CBS-TA and ECBS-TA can not plan for the $250 \times 250$ sized maze.

---

[2]https://github.com/whoenig/libMultiRobotPlanning

# F    Evaluation of Learning Ability

Table 8: Graph dataset details for graph node classification.

| Name | #Nodes | #Edges | Metric |
|------|--------|--------|--------|
| Cora | 2,708 | 5,429 | Accuracy |
| CiteSeer | 3,327 | 4,732 | Accuracy |
| PubMed | 19,717 | 44,338 | Accuracy |
| ogbn-arxiv | 169,343 | 1,166,243 | Accuracy |
| ogbn-products | 2,449,029 | 61,859,140 | Accuracy |
| ogbn-proteins | 132,534 | 39,561,252 | ROC-AUC |

We have evaluated the planning ability of our GPFNN in the previous section. In this section, we further evaluate the learning ability of our GPFNN on the graph node classification tasks. The goal is to show that our GPFFN has the learning ability and, thus, it may have the potential to handle tasks that require both learning ability and planning ability. We employ simple three-layer neural networks to learn the initial potential and visual node potential in our GPFNN.

We first evaluate our GPFNN on three large graph datasets. Namely, the ogbn-arxiv, ogbn-products, and ogbn-proteins graph datasets. The ogbn-arxiv dataset represents a citation network between all computer science papers on arXiv, where each node represents a paper and each edge represents a citation between papers. The ogbn-products dataset is an Amazon product co-purchasing network, where each node represents a product sold on Amazon, and each edge represents that the products are purchased together. The ogbn-proteins dataset is an undirected graph, where each node represents a protein, and each edge indicates different types of biologically meaningful associations between proteins.

To evaluate our GPFNN, we compare with GCN (Kipf & Welling, 2017), DeeperGCN (Li et al., 2020a), and RevGNN (Li et al., 2021). The classification accuracy and the number of parameters are shown in Table 9. We report the classification accuracy and the number of parameters of GCN, DeeperGCN and RevGNN from their paper. From Table 9, we can observe that GPFNN obtains a consistently higher classification accuracy and a smaller number of parameters compared with GCN. Moreover, GPFNN achieves a competitive classification accuracy on ogbn-products and ogbn-arxiv datasets, while it has a very small number of parameters compared with RevGNN. In addition, although GPFNN has a significantly smaller number of parameters compared with DeeperGCN, it obtains a higher classification accuracy on ogbn-products and ogbn-arxiv than DeeperGCN.

The experiments of node classification on large datasets show the potential of GPFNN on learning tasks even beyond the primary focus of GPFNN on graph path planning tasks. It is interesting to extend GPFNN on multi-task learning that supports multi-agent path planning and node action prediction (node classification). Such extension may naturally form a backbone for multi-agent reinforcement learning in tasks requiring path planning and cooperation, e.g., multi-agent RL in city package delivery, multi-agent games, etc. We leave this interesting application as one of our future works.

Table 9: Node Classification Accuracy % (ROC-AUC) and # Parameter of Model

|  |  | ogbn-arxiv | ogbn-products | ogbn-proteins |
|------|------|------------|---------------|---------------|
| GPFNN | Acc | $72.19 \pm 0.14$ | $81.70 \pm 0.42$ | $74.45 \pm 0.17$ |
|  | #Parameter | **22780** | **20106** | **13756** |
| GCN (Kipf & Welling, 2017) | Acc | $71.74 \pm 0.29$ | $75.64 \pm 0.21$ | $72.51 \pm 0.35$ |
|  | #Parameter | 110120 | 103727 | 96880 |
| DeeperGNN (Li et al., 2020a) | Acc | $71.92 \pm 0.16$ | $80.98 \pm 0.20$ | $85.80 \pm 0.17$ |
|  | #Parameter | 491176 | 253743 | 2374568 |
| RevGNN (Li et al., 2021) | Acc | **$73.01 \pm 0.31$** | **$83.07 \pm 0.30$** | **$87.74 \pm 0.13$** |
|  | #Parameter | 262000 | 2945007 | 20031384 |

Table 10: Summary of classification accuracy (%) results with various depths.

| Dataset | Method | Layers | | | | | | Std |
|---|---|---|---|---|---|---|---|---|
| | | 2 | 4 | 8 | 16 | 32 | 64 | |
| Cora | GCN | 81.1 | 80.4 | 69.5 | 64.9 | 60.3 | 28.7 | 19.25 |
| | JKNet | - | 80.2 | 80.7 | 80.2 | 81.1 | 71.5 | 4.06 |
| | Incep | - | 77.6 | 76.5 | 81.7 | 81.7 | 80.0 | 2.37 |
| | GCNII* | 80.2 | 82.3 | 82.8 | 83.5 | 84.9 | 85.3 | 1.86 |
| | GPFNN | 82.46 | 82.31 | 83.20 | 83.20 | 82.81 | 83.38 | **0.44** |
| Citeseer | GCN | 70.8 | 67.6 | 30.2 | 18.3 | 25.0 | 20.0 | 24.05 |
| | JKNet | - | 68.7 | 67.7 | 69.8 | 68.2 | 63.4 | 2.45 |
| | Incep | - | 69.3 | 68.4 | 70.2 | 68.0 | 67.5 | 1.08 |
| | GCNII* | 66.1 | 67.9 | 70.6 | 72.0 | 73.2 | 73.1 | 2.91 |
| | GPFNN | 69.88 | 70.46 | 70.58 | 70.37 | 70.44 | 70.9 | **0.33** |
| Pubmed | GCN | 79.0 | 76.5 | 61.2 | 40.9 | 22.4 | 35.3 | 23.30 |
| | JKNet | - | 78.0 | 78.1 | 72.6 | 72.4 | 74.5 | 2.80 |
| | Incep | - | 77.7 | 77.9 | 74.9 | OOM | OOM | 1.68 |
| | GCNII* | 77.7 | 78.2 | 78.8 | 80.3 | 79.8 | 80.1 | 1.07 |
| | GPFNN | 78.87 | 80.23 | 79.54 | 79.09 | 79.31 | 79.09 | **0.49** |

We further evaluate the effect of the number of layers of GPFNN on three small graph datasets. Specifically, the Cora, CiteSeer and PubMed (Sen et al., 2008) are employed. Cora, CiteSeer and PubMed (Sen et al., 2008) are citation networks, where each node is a paper with bag-of-words attributes, and each edge indicates the citation relationship between papers. All the details of datasets employed are summarized in Table 8.

We compare GPFNN with GCN (Kipf & Welling, 2017), JKNet (Xu et al., 2018) IncepGCN (Rong et al., 2020) and GCNII (Chen et al., 2020). The number of layers in the empirical studies are set to $\{2, 4, 8, 16, 32, 64\}$. The detailed classification accuracy of different methods with different layers are shown in Table 10. From Table 10, we can observe that the classification accuracy of GCN and JKNet drop as the depth increases. Moreover, we can see that GPFNN achieves a competitive classification accuracy compared with the state-of-the-art method GCNII. To measure the stability of each method over different depths of models, the standard deviation of the classification accuracy over different layers is shown in the last column in Table 10. It's easy to find that the performance of GPFNN is more stable than all baselines when the depth increases.

# G  Additional Related Works about Multi-agent Reinforcement Learning

Recently, several deep multi-agent reinforcement learning (MARL) methods are proposed to cooperative control of multi agent systems. In (Lowe et al., 2017), the authors propose a deep reinforcement learning algorithm called MAAC for multi-agent pathfinding (MAPF) in mixed cooperative-competitive settings. (Foerster et al., 2016) proposes a novel approach for multi-agent reinforcement learning by introducing communication between agents using a continuous message passing mechanism. The authors introduce a new architecture called CommNet, which consists of a shared policy network and a communication module that enables agents to send and receive messages.

Recent work has explored using deep multi-agent reinforcement learning to learn policies for solving MAPF. In (Sartoretti et al., 2019) the authors propose a method for multi-agent pathfinding (MAPF) that combines reinforcement learning and imitation learning. The authors introduce a new algorithm called PRIMAL, which leverages both individual and collective rewards to learn effective policies for MAPF. The PRIMAL algorithm consists of two phases: a reinforcement learning phase and an imitation learning phase. In the reinforcement learning phase, each agent learns an individual policy that maximizes its own reward. In the imitation learning phase, agents learn a collective policy by imitating the actions of an expert agent, which is a centralized planner that produces paths for all agents. However, it can not scale to large sized problems. In addition, it can not deal with dynamic targets.

In (Ma et al., 2021), the authors propose a distributed heuristic approach to the (MAPF) problem. The approach is based on a centralized search that is distributed among the agents, who communicate with each other to coordinate their actions. The proposed algorithm consists of two main phases: a communication phase and a planning phase. In the communication phase, each agent broadcasts its current state and receives the states of its neighbors. Based on this information, the agents construct a communication graph that reflects the current state of the environment. In the planning phase, each agent computes a local plan using a heuristic search algorithm that takes into account the current communication graph. Although the proposed method improves the scalability compared with PRIMAL, the scale of the problems is still limited. Most recently, some works (van Knippenberg et al., 2021) try to employ MARL for time-Constrained MAPF on general graphs. Recent work (Li et al., 2022) combines MARL with a proposed Prioritized Communication Learning framework that uses a centralized communication network to facilitate communication between agents. The agents prioritize their communication channels based on the current state of the environment, and use the prioritized communication channels to exchange information about their current plans and goals.

Although Deep MARL methods are general and flexible to handle different tasks including MAPF problems, they suffer from scalability issue that arises when dealing with large numbers of agents. As the number of agents increases, the complexity of the joint action space grows exponentially, making it challenging to find good solutions. Moreover, current MARL methods often suffer from slow learning rates and poor sample efficiency, which can limit their applicability in real-world settings.

