# OpenReview forum: "Graph Potential Field  Neural Network for  Massive Agents Group-wise  Path Planning"
_TMLR — Accepted by TMLR_

### Review · Reviewer_DF4B · 2024-12-23

**Summary Of Contributions:**

This paper presents a GNN-based approach to single-group target assignment and pathfinding tasks (TAPF, also known as anonymous/unlabeled MAPF). The work has practical applications in warehouse automation, where robots need to efficiently move items regardless of specific agent-goal combinations. The proposed GPFNN seems to comprise two main networks: a GCN to sequentially convolute a given maze's graph structure and construct a potential field, and a collision-aware neural decoder that processes this potential field to generate valid actions, both for all agents. The authors evaluate GPFNN against a few widely known algorithms used for the classic MAPF task.

**Audience:**

Yes

**Claims And Evidence:**

No

**Requested Changes:**

Although I admire the idea of solving MAPF using a GNN, I believe this paper needs a fundamental rewrite to be accepted.

- [critical] Conduct a comprehensive literature review around TAPF
- [critical] Clarify the argument structure
- [critical] Implement fair evaluation in accordance with existing works
- [critical] Improve presentation & writing
    - It would also be helpful visualizing the solving process e.g., a tiny maze with two agents going home

**Strengths And Weaknesses:**

Strengths:

- The paper offers an interesting perspective on the MAPF task by treating it as a graph of paths with obstacles and moving agents, effectively leveraging GNN architecture to exploit potential field characteristics for given problem states.

Weaknesses:

- The authors demonstrate insufficient awareness of the TAPF variant, despite its critical relevance. While "target assignment" is briefly mentioned (pp. 1-2), the paper fails to acknowledge significant prior work, including the optimal algorithm [0], CBM [1], CBS-TA [2], and TSWAP [3].
- Argument
    - Lists “learning ability” as part of contribution, but the experiments are not in the body but in one of Appendices.
    - The assumption regarding potential fields is overly optimistic. The paper incorrectly suggests that valid potential fields guarantee monotonic flow and optimal paths, when in fact:
        - Monotonicity is not guaranteed due to local minima and obstacles
        - Potential fields only ensure feasible paths, not optimal or monotonic ones
        - Even the cited Masoud (2008) paper does not support the claim about potential fields avoiding local minima
    - There is no justification given as to why we would expect the *collision-free* network to perform well on other tasks like node classification.
- Evaluation
    - Eval is unfair, comparing the proposed method to the algorithms specifically written to solve *classic* MAPF when `libMultiRobotPlanning` clearly has task-assignment variants.
    - With node classification as well, the quantitative comparisons aren’t fair either unless the trade-off with model size is justified; given a fixed amount of data, the performance strongly depends on model size (scaling laws).
- Presentation
    - It took some time just to know that this paper is about TAPF, which I could barely notice before skimming the Supplementary material
    - Paper flow is confusing. Section 4 is devoted to graph learning represented by node classification which isn’t addressed until Appendix E and not the main point of the paper. Also, the abbreviations IPNN and VNPNN are **not** used in the Appendix.
    - Figures 1 and 4: presenting mazes with exit goal is not very representative of MAPF; the potential field would be better with a random goal within the maze
    - Redundantly duplicate contents in 4.1.1 and 4.1.2. Just describe the specification
    - Many uses of `\citep` when `\citet` is appropriate
    - Too many “[~]ing ability of our GPFNN” (x12?)
    - 2nd column of Table 1 is missing its name (No. of agents)
    - Numerous writing errors/typos.
- Confusions
    - I am confused by the very first sentence of abstract. Path planning and pathfinding are pretty much the same thing, and MARL is a method to tackle it.

        > Multi-agent path planning is important in both multi-agent path finding and multi-agent
        reinforcement learning areas.
        >
    - Nowhere in the manuscript is it written the training details about action decoder: data, batch size, learning rate, optimizer, etc.
    - If architecturally designed to avoid collisions, why would it fail in some cases?
    - EPEA* is first introduced as one of multi-agent solvers just to be rejected 2 paragraphs later?
    - It is not clear even whether the potential field network itself is optimized before or together with the action decoder. Or is it that message passing inside the GNN is expected to propagate something like state values from goals?
    - What do you mean by a “visual node”?
    - LSE doesn’t sound necessary when feature values are expected to vary to some extent. Wray, et al. (2016) only suggested the use of LSE for there being a precision issue around near-1 floating points. The standard average pooling feels right to me for this purpose.
    - As in Section 4.2, is IPNN used only as a regularizer in Appendix E? You could use KL Divergence or something instead of MSE.
    - Both section 4 and Appendix E alltogether, but it’s also not fair highlighting #Parameter of GPFNN when GCN is smaller.

---

[0] Yu, Jingjin, and Steven M. LaValle. "Multi-agent path planning and network flow." Algorithmic Foundations of Robotics X: Proceedings of the Tenth Workshop on the Algorithmic Foundations of Robotics. Berlin, Heidelberg: Springer Berlin Heidelberg, 2013.

[1] Ma, Hang, and Sven Koenig. "Optimal target assignment and path finding for teams of agents." arXiv preprint arXiv:1612.05693 (2016).

[2] Hönig, Wolfgang, et al. "Conflict-based search with optimal task assignment." Proceedings of the International Joint Conference on Autonomous Agents and Multiagent Systems. 2018.

[3] Okumura, Keisuke, and Xavier Défago. "Solving simultaneous target assignment and path planning efficiently with time-independent execution." Artificial Intelligence 321 (2023): 103946.

---

> ### Author Response · Authors · 2025-01-27
>
> Thanks for the reviewer’s detailed comments.  We respectfully disagree with part of the reviewer’s argument. We would like to clarify the potential misunderstanding. Detailed responses to the reviewer's questions are given below.
>
>
> **W1** “The assumption regarding potential fields is overly optimistic. The paper incorrectly suggests that valid potential fields guarantee monotonic flow and optimal paths, when in fact:
> 1.	Monotonicity is not guaranteed due to local minima and obstacles
> 2.	Potential fields only ensure feasible paths, not optimal or monotonic ones
> 3.	Even the cited Masoud (2008) paper does not support the claim about potential fields avoiding local minima."
>
> **Q1:** "Monotonicity is not guaranteed due to local minima and obstacles"
>
> **A1:** The Potential fields that satisfy the Laplace’s equation does not have local minima inside the domain [1]. One famous example that satisfy the Laplace’s equation is the harmonic function. It does not have local minima in the interior of the domain.  It has extremes only on the boundary of the domain [2].  Because of this nice property of the harmonic function,  the harmonic potential field have been employed for path planning [2,3, 4, 5, 6].  Masoud (2008) paper employs the graph potential field. Our work builds upon  Masoud (2008) that satisfy the graph Laplace’s equation, which does not have local minima.
>
> **Q2:** “Potential fields only ensure feasible paths, not optimal or monotonic ones”
>
> **A2:** Graph potential field satisfies the Laplace’s equation. Setting the boundary condition of obstacles to a large value b and the target to a small value, e.g, 0.  Because the graph potential field does not have local minima in the interior of the domain,  and the minima are only on the boundary of the domain,  there is a path from any node to the target node (boundary) that monotonically decreases the potential.  When employing the Graph potential field to perform path planning, one can greedily select nodes by fastest potential descent. Because there are no local minima and the potential monotonically decreasing, the greedy selected path is the optimal path that decreases the potential in the fastest way.
>
>
> **Q3:**  “Evaluation is unfair, comparing the proposed method to the algorithms specifically written to solve classic MAPF when libMultiRobotPlanning clearly has task-assignment variants.”
>
> **A3:** The TAPF problems studied in the listed papers including the optimal algorithm [Yu et al. 2013], CBM [Ma et al. 2016], CBS-TA [Hönig et al 2018], and TSWAP [Keisuke et al. 2023] are different from the problem we focus on.
>
> The TAPF problem they studied requires assigning each agent a unique target (location). In contrast,  our focused group-wise path planning problem does not have this assumption. Agents that achieve (reach) the target can continue to search for the remaining targets.
>
> Nevertheless, we evaluate the CBS-TA and ECBS-TA in libMultiRobotPlanning on 129x129 mazes and 401x401 mazes with 100 agents. CBS-TA and ECBS-TA timeout and fail to output paths in 72000 seconds. We provide all the mazes in the supplementary material.
>
> **Q4:** “With node classification as well, the quantitative comparisons aren’t fair either unless the trade-off with model size is justified; given a fixed amount of data, the performance strongly depends on model size (scaling laws).”
>
> **A4:** Both the model size of baselines and datasets are the same as the baselines’ papers.  Our GPFNN achieves a competitive performance with a significantly smaller model size compared with baselines.
>
> **Q5:**  “If architecturally designed to avoid collisions, why would it fail in some cases?”
>
> **A5:** We guess the reviewer means that the success rate of GPFNN is not 100% in experiments. The success rate is not 100% is not because of the collisions. It is because the GPFNN reaches the maximum moving steps and there are still some targets not been reached.  This may happen when some targets have been randomly initialized in disconnected components with agents, especially for complex mazes with high obstacle densities.   The success rate of CBS-TA on 8x8 maps is not always 100% as shown in their paper [Hönig et al 2018].
>
>
>
>
>
> 	[1] C. I. Connolly, J. B. Burns, and R. Weiss. Path planning using Laplace’s equation. In Proceedings of the IEEE International Conference on Robotics and Automation, pages 2102–2106, 1990.
>
> 	[2] Szulczyński et al.  REAL-TIME OBSTACLE AVOIDANCE USING HARMONIC POTENTIAL FUNCTIONS. Journal of Automation, Mobile Robotics & Intelligent Systems. 2011.
>
> 	[3]  Daily et al.  Harmonic Potential Field Path Planning for High Speed Vehicles. American Control Conference. 2008.
>
> 	[4] Panati et al.  Autonomous Mobile Robot Navigation Using Harmonic Potential Field. 2015
>
> 	[5] Kim et al. Real-Time Obstacle Avoidance Using Harmonic Potential Functions. IEEE Transaction on Robotics Automation. 1992.
>
> 	[6] Wang et al.  A new potential field method for robot path planning.  2000

---

> > ### Comment · Reviewer_DF4B · 2025-01-29
> >
> > Thank you for your response. While I appreciate your efforts in addressing part of my concerns, several key issues remain unresolved. To keep the discussion focused, I will highlight the most critical ones below.
> >
> > First, I apologize for any initial misunderstanding of your approach to multi-agent path planning. As I now understand it, you are tackling a **further relaxed variant of TAPF**—one in which some agents reach multiple goals while others reach none, as long as all goals are eventually covered. Given this, I have additional concerns as follows:
> >
> > 1. The authors stated that they did not compare their approach to TAPF algorithms. However, the same reasoning could be applied to MAPF, which features a stricter pre-assignment structure. If TAPF methods were excluded from comparison, how do the authors justify benchmarking against MAPF methods instead?
> > 2. In response to my concerns, the authors tested a couple of TA methods, both of which failed:
> >     - It is particularly odd that ECBS with pre-assignment performed to a certain degree while more relaxed TA methods did not. What might explain this discrepancy?
> >     - There is a chance that the evaluated mazes are ill-posed, such that some agents are trapped in subspaces and unable to navigate to their goals, significantly impacting the performance of existing methods. Could the authors confirm:
> >         1. That valid paths exist for all agents in every maze?
> >         2. That TA methods are capable of solving small-scale mazes?
> > 3. More broadly, I wonder why the authors would not evaluate GPFNN as a TAPF method. It feels as though the problem setup is framed to leverage the potential field to guide agents based on proximity to their nearest targets. However, it should be entirely feasible to stop agents once they reach any goal, letting the remaining agents complete the task. This would make the comparison fair.
> >
> > I remain open to reconsidering my recommendation if the above concerns, along with the following unresolved issues, are properly addressed:
> >
> > **Major**
> >
> > - Nowhere in the manuscript is it written the training details about action decoder: data, batch size, learning rate, optimizer, etc.
> > - It is not clear even whether the potential field network itself is optimized before or together with the action decoder. Or is it that message passing inside the GNN is expected to propagate something like state values from goals?
> > - Lists “learning ability” as part of contribution, but the experiments are not in the body but in one of Appendices.
> > - Redundantly duplicate contents in 4.1.1 and 4.1.2. Just describe the specification
> >
> > **Minor**
> >
> > - There is no justification given as to why we would expect the collision-free network to perform well on other tasks like node classification.
> > - Paper flow is confusing. Section 4 is devoted to graph learning represented by node classification which isn’t addressed until Appendix E and not the main point of the paper. Also, the abbreviations IPNN and VNPNN are not used in the Appendix.
> >
> > ---
> >
> > As a side note, the problem setup also relates to the lifelong variant of MAPF, in which agents are assigned to another target as soon as they reach one.

---

> ### Author Response · Authors · 2025-03-19
>
> We sincerely appreciate the reviewer's detailed comments and valuable suggestions.  Detailed responses to the reviewer's questions are given below.
>
>
> **Q1:** “The authors stated that they did not compare their approach to TAPF algorithms. However, the same reasoning could be applied to MAPF, which features a stricter pre-assignment structure. If TAPF methods were excluded from comparison, how do the authors justify benchmarking against MAPF methods instead?”
>
> **A1:** We test the baseline algorithms by employing the code in the  libMultiRobotPlanning package.  The MAPF methods ( ECBS, and P-Planning) can output trajectory files (for part of the agents) despite their failure to plan all agent-target pairs.  These results support computing the mean score metric and the mean finishing steps metric of the baselines.    In contrast,   the TAPF methods (CBS-TA and ECBS-TA) can not output any trajectory files and fail to plan on all the cases.
>
>
>
> **Q2.** “It is particularly odd that ECBS with pre-assignment performed to a certain degree while more relaxed TA methods did not. What might explain this discrepancy?”
>
> **A2:** Both ECBS and ECBS-TA fail to path plan for all agent-target pairs.  The difference is that ECBS in libMultiRobotPlanning output path for part of the agent-target pairs.   In contrast, ECBS-TA  in libMultiRobotPlanning did not output any files and time out on all the cases.
>
> **Q3:** “There is a chance that the evaluated mazes are ill-posed, such that some agents are trapped in subspaces and unable to navigate to their goals, significantly impacting the performance of existing methods.”
>
> **A3:**  The mazes employed  have  dense obstacles and narrow roads, which are very challenging  There is a chance that some agents (randomly initialized) are trapped in disconnected components of the graph.  However,  both our GPFNN and baselines are evaluated on the same mazes. The experiments at least show that the baselines can not handle our planning problems, while our GPFNN performs well.   We have provided all the mazes in the supplement material.
>
>
> **Q4:** “More broadly, I wonder why the authors would not evaluate GPFNN as a TAPF method. It feels as though the problem setup is framed to leverage the potential field to guide agents based on proximity to their nearest targets. However, it should be entirely feasible to stop agents once they reach any goal, letting the remaining agents complete the task. This would make the comparison fair.”
>
> **A4:**  Thanks for the reviewer’s suggestion. We further evaluate GPFNN in the TAPF setting.  More details can be found in the section E on page 20 of the Appendix. We compare GPFNN with CBS-TA and ECBS-TA on 54x54-sized maze and 250x250-sized maze with wider roads. The experimental results show our GPFNN can achieve consistently higher success rates compared with CBS-TA and ECBS-TA.

---

### Review · Reviewer_1xdK · 2024-12-26

**Summary Of Contributions:**

The paper proposes a Graph Potential Field Neural Network (GPFNN) for group-wise multi-agent path planning, addressing scalability and over-smoothing issues in existing methods. It incorporates dynamic boundary conditions and demonstrates improved performance in controlled scenarios.

**Audience:**

Yes

**Claims And Evidence:**

Yes

**Requested Changes:**

1. Please see the Weaknesses above. I would expect these points to be more clearly addressed in the conclusion.

2. Add a Limitations sections where limitations such as computational costs and assumptions on agents are more clearly summarized

**Strengths And Weaknesses:**

## Strengths:

1. The GPFNN introduces a novel approach by modeling potential fields as a neural network, effectively addressing over-smoothing issues and enabling scalable optimization.

2. Demonstrated success on large-scale problems, including mazes up to 1025 × 1025 and tasks involving up to 1,000 agents, highlights its capacity to handle complex and large-scale environments.

3. Extensive experiments with multiple baselines show the method's superiority in terms of planning efficiency, success rates, and finishing time across various test cases.

## Weaknesses:

1. The experiments are confined to controlled settings, leaving its practical applicability in real-world, heterogeneous, and dynamic environments uncertain.

2. The computational cost of the GPFNN's deeper architectures and multi-agent simulations is not clearly addressed, which might pose challenges for real-time applications.

3. The method assumes uniform agent capabilities and objectives, which might limit its effectiveness in scenarios with heterogeneous agents or complex cooperation dynamics.

---

> ### Author Response · Authors · 2025-01-27
>
> We sincerely appreciate the reviewer's constructive advice and valuable comments.  Detailed responses to the reviewer's questions are given below.
>
> **Q1:** “The experiments are confined to controlled settings, leaving its practical applicability in real-world, heterogeneous, and dynamic environments uncertain.” “The method assumes uniform agent capabilities and objectives, which might limit its effectiveness in scenarios with heterogeneous agents or complex cooperation dynamics”
>
>
> **A1:**  Thank you for the insightful comments. Currently, our work focuses on path planning for homogeneous agents. Path planning for multiple heterogeneous agents presents a more challenging problem that needs further investigation. One potential approach is to adopt a hierarchical planning strategy. At the high level, our GPFNN could be utilized to perform abstract planning in a heterogeneous way, while at the low level, high-resolution grids could be employed to perform more precise, high-resolution planning tailored to the specific characteristics of the heterogeneous agents. We include a limitation section in our revised paper.
>
> **Q2:** “The computational cost of the GPFNN's deeper architectures and multi-agent simulations is not clearly addressed, which might pose challenges for real-time applications.”
>
> **A2:**  .  In terms of computational complexity,  the computation of the graph potential field in our GPFNN can be performed by graph message passing, which can be computed in parallel by GPUs with the same complexity as standard GNNs. The difference is that the depth in GPFNN determines the potential field accuracy. A deeper GPFNN can achieve higher potential field accuracy, and lead to a higher computation cost.

---

### Review · Reviewer_h2q8 · 2025-01-13

**Summary Of Contributions:**

In this paper, the authors introduce a method to tackle the multi-agent pathfinding problem (MAPF). Specifically, they draw inspiration from the graph potential field to formulate the problem. The main contribution lies in the idea of adding virtual nodes as boundary conditions to avoid trivial solutions and expand the model's capacity to work in dynamic settings. The authors also provide a theoretical analysis of the advantages of their solution and present empirical experiments to highlight the model's performance.

**Audience:**

Yes

**Claims And Evidence:**

Yes

**Requested Changes:**

Please refer to the Weakness.

**Strengths And Weaknesses:**

## Strengths:
- The paper is well-structured, with clear sections for theory, architecture, and experiments.
- The addition of virtual nodes is a simple yet effective technique. It addresses key challenges in MAPF, including. 1. Over-smoothing when the problem is converted to a graph neural network framework, 2. Dynamic weighting in MAPF scenarios.
- The empirical results demonstrate the method's potential, showcasing strong performance on small—and large-scale problems across diverse metrics.

## Weaknesses:
- The paper does not delve into how GPFNN compares against state-of-the-art GNN-based planners in terms of computational complexity, scalability, or performance. A clearer discussion or quantitative comparison would strengthen the argument for the proposed method's advantages
- Evaluations on real-world datasets such as multi-robot path planning in warehouses or traffic management scenarios could validate the method's applicability beyond synthetic mazes.

---

> ### Author Response · Authors · 2025-01-27
>
> We sincerely appreciate the reviewer's constructive advice and valuable comments.  Detailed responses to the reviewer's questions are given below.
>
> **Q1:** “How GPFNN compares against state-of-the-art GNN-based planners in terms of computational complexity, scalability, or performance.”
>
> **A1:** The current GNN-based planners [1-4] focus on the MAPF problems. Unlike these methods, our work is focused on group-wise MAPF problems. Additionally, our GPFNN is capable of handling dynamic targets, whereas existing GNN-based methods are limited to static targets. Furthermore, our GPFNN can scale to highly complex 1025x1025 mazes with dense obstacle configurations, a level of scalability that the experiments in these prior works have not achieved.  In terms of computational complexity,  the computation of graph potential field in our GPFNN can be performed by graph message passing, which can be computed in parallels by GPUs with the same complexity as standard GNNs. The difference is that the depth in GPFNN determines the potential field accuracy. A deeper GPFNN can achieve higher potential field accuracy, and lead to a higher computation cost.
>
>
> **Q2:** “Evaluations on real-world datasets such as multi-robot path planning in warehouses or traffic management scenarios could validate the method's applicability beyond synthetic mazes.”
>
> **A2:**  Thanks for the reviewer’s comments.  Currently, we evaluate our GPFNN on complex and large-scale mazes with high obstacle densities.  These mazes are more complex than traffic networks, which is more challenging for multi-agent path planning. To test the actual planning abilities, a complex one is more suitable than an easy one (e.g., maps with only a few obstacles). We provide all the mazes in the supplementary material.
>
> [1] Qingbiao Li, Fernando Gama, Alejandro Ribeiro, and Amanda Prorok. Graph neural networks for decentralized multi-robot path planning. 2020 IEEE/RSJ International Conference on Intelligent Robots and Systems (IROS), pp. 11785–11792, 2019.
>
> [2] Qingbiao Li, Weizhe Lin, Zhe Liu, and Amanda Prorok. Message-aware graph attention networks for
> large-scale multi-robot path planning. IEEE Robotics and Automation Letters, 6:5533–5540, 2020
>
> [3] Hanqi Dai, Weining Lu, Jun Yang, and Bin Liang.  Cooperative path planning of multi-agent based on graph neural network. In 2022 34th Chinese Control and Decision Conference (CCDC), pp. 5620–5624, 2022
>
> [4] Dongming Zhou, Zhengbin Pang, and Wei Li. Enhanced causal reasoning and graph networks for multi-agent path finding. In 2024 International Joint Conference on Neural Networks (IJCNN), pp. 1–8, 2024

---

### Decision · Action_Editor_9Abc · 2025-02-23

**Recommendation:** Accept with minor revision

**Comment:**

The major concern was about the problem setup, e.g., the lack of clarity about the difference between MAPF and TAPF settings or why the authors did not evaluate GPFNN on TAPF tasks. I suggest the authors to invest more time on alleviating this issue for their final submission. It would be great if the authors could evaluate GPFNN for the TAPF setup too.

However, I do not think the concerns invalidate the soundness of the approach. I would like to recommend acceptance since this is an interesting submission with ideas from potential-based path planning algorithms. I believe the work is solid and convincing from the perspective of both theory and experiments.

**Audience:**

Yes, there are plenty of researchers who are interested in graph neural networks, planning, and deep neural network-based algorithmic reasoning.

**Claims And Evidence:**

The authors provide sufficient evidence for their claims through experiments and surveys on related works.

The claims are as follows:
- The authors are the first to develop graph potential field-based neural network with planning ability.
- The proposed algorithm works for both graph path planning tasks and graph learning tasks
- The proposed algorithm works better than the considered baselines even for challenging experiments (1025x1025 grid).

---

> ### Author Response · Authors · 2025-03-19
>
> We sincerely thank the AE and all reviewers for their great efforts and valuable comments.
>
> We have revised the paper according to the AE and reviewers’ suggestions.
>
> - We highlight the difference between our focused problem and the TAPF in the blue paragraph on page 3 of our revised paper.   The TAPF setting requires the agent to stop once it reaches the target. In contrast, we focus on the continual group-wise planning problems in which the agents that achieve (reach) the target can continue to search for the remaining targets.  Moreover,  our method supports handling dynamic moving targets, while most TAPF methods cannot.
>
>
> - We further evaluate our GPFNN in the TAPF setting as the AE and reviewers’ suggested.  More details can be found in the section E on page 20 of the Appendix.  We compared GPFNN with the TAPF baselines (CBS-TA and ECBS-TA) on 54x54-sized maze and 250x250-sized maze.   The experimental results show our GPFNN can achieve consistently higher success rates compared with the baselines in the TAPF setting.